# Time-gated detection of protein-protein interactions with transcriptional readout

**Min Woo Kim**[1†]**, Wenjing Wang**[1†]**, Mateo I Sanchez**[1]**, Robert Coukos**[1]**, Mark von Zastrow**[2]**, Alice Y Ting**[1,3,4,5‡]*

[1]Department of Genetics, Stanford University, Stanford, United States; [2]Program in Cell Biology, University of California, San Francisco, San Francisco, United States; [3]Department of Biology, Stanford University, Stanford, United States; [4]Department of Chemistry, Stanford University, Stanford, United States; [5]Chan Zuckerberg Biohub, San Francisco, United States

**Abstract** Transcriptional assays, such as yeast two-hybrid and TANGO, that convert transient protein-protein interactions (PPIs) into stable expression of transgenes are powerful tools for PPI discovery, screens, and analysis of cell populations. However, such assays often have high background and lose information about PPI dynamics. We have developed SPARK (Specific Protein Association tool giving transcriptional Readout with rapid Kinetics), in which proteolytic release of a membrane-tethered transcription factor (TF) requires *both* a PPI to deliver a protease proximal to its cleavage peptide *and* blue light to uncage the cleavage site. SPARK was used to detect 12 different PPIs in mammalian cells, with 5 min temporal resolution and signal ratios up to 37. By shifting the light window, we could reconstruct PPI time-courses. Combined with FACS, SPARK enabled 51 fold enrichment of PPI-positive over PPI-negative cells. Due to its high specificity and sensitivity, SPARK has the potential to advance PPI analysis and discovery.
DOI: https://doi.org/10.7554/eLife.30233.001

*For correspondence:
ayting@stanford.edu

†These authors contributed equally to this work
‡Alice Y. Ting is the corresponding author.

## Introduction

Protein-protein interactions (PPIs) are central to cellular signal transduction. Consequently, many assays have been developed to detect and study them, particularly in the context of living cells, where native PPIs are unperturbed by cell lysis, detergents, fixatives, or dilution. For example, FRET (*Truong and Ikura, 2001*), BRET (*Pfleger and Eidne, 2006*), fluorescence correlation spectroscopy (FCS) (*Krichevsky and Bonnet, 2002*), protein complementation assays (PCAs) (*Shekhawat and Ghosh, 2011*), and fluorescence relocalization assays (*Teruel and Meyer, 2000*; *Yang et al., 2013*) have all been applied to visualize PPI dynamics in living cells. Though information-rich, these assays have the downside of being labor-intensive, requiring high-content or time-lapse microscopy, and are consequently difficult to perform on a large scale. This makes them non-optimal for PPI discovery, for adaptation to high-throughput screens, or for analysis of large cell populations such as those found within complex tissue. Instead, real-time assays are better suited for the focused study of a small number of known PPIs under a small set of conditions, or in a small number of cells.

A separate class of assays detects PPIs by signal integration rather than real-time imaging, and produces gene transcription as the readout. Examples include the yeast two hybrid assay (*Miller and Stagljar, 2004*), the split ubiquitin assay (*Petschnigg et al., 2014*), and TANGO (*Barnea et al., 2008*; *Inagaki et al., 2012*; *Kroeze et al., 2015*). Benefits of these assays include scalability (because real-time microscopy is not needed), versatility of read out (transcription of a fluorescent protein or an antibiotic resistance gene, for example), and high sensitivity due to signal amplification. These properties have led to the widespread use of integrative PPI assays for PPI discovery and drug screening. However, these assays have two major drawbacks. First, because they

integrate PPI events over long time periods, typically 18 hr to days (i.e., over the entire time that the tools are expressed in cells), they often produce high background, leading to high rates of false discovery. Second, these assays lose information about PPI dynamics. It is impossible to know, for instance, whether a PPI event occurred during a specific 5 min time window of interest or at another time during the tool expression window.

To address these limitations, and to expand the utility and versatility of PPI assays that work by signal integration and transcriptional readout, we report a new tool called SPARK, for 'Specific Protein Association tool giving transcriptional Readout with rapid Kinetics'. SPARK detects interactions in living cells between specific protein pairs of interest (proteins A and B in *Figure 1A*), and produces gene transcription as a result. In contrast to previous tools, SPARK is also gated by externally-applied blue light. Hence, transcriptional activation requires both protein A-protein B interaction *and* light, the latter of which can be applied during any user-specified time window. This generic and non-invasive form of temporal gating enables SPARK to capture PPI dynamics to some extent, and reduces background signal overall, while preserving the tremendous benefits of transcriptional readout.

Here we describe the development of SPARK, its characterization in living mammalian cells, and its application to a range of PPIs, including eight different GPCRs. We use SPARK to compare the temporal profiles of different GPCRs interacting with β-arrestin2 in response to various ligands. Finally, we show that SPARK can enable FACS-based enrichment of cells that experienced specific transient PPI events >9 hr earlier. This lays the foundation for the eventual use of SPARK for genetic screens and PPI discovery on a genome-wide scale.

## Design and optimization of SPARK

To design SPARK, we built upon our recently reported FLARE tool, which is a transcription factor gated by the coincidence of light and high calcium (*Wang et al., 2017*). As shown in *Figure 1A–B*, SPARK has three components: a transcription factor (TF) that is tethered to protein interaction partner 'A' as well as to the plasma membrane (or other intracellular membrane, sequestered from the nucleus) via a protease cleavage site ('TEVcs' for tobacco etch virus protease cleavage site); a protease ('TEVp' for tobacco etch virus protease) that is fused to protein interaction partner 'B'; and a reporter gene whose transcription is triggered by TF translocation to the nucleus. TF release from the plasma membrane via TEVcs cleavage requires the coincidence of two events: protein A-protein B interaction *and* externally-applied blue light. This is because, first, the TEVp protease is tuned to have low affinity for its TEVcs substrate and cleavage is only significant when the two are brought into proximity via A-B interaction (*Wang et al., 2017*). Second, the TEVcs is caged by an 'evolved LOV domain' (eLOV), which we previously generated by directed evolution to tightly cage the TEVcs sequence and prevent its cleavage in the dark state (*Wang et al., 2017*). A brief (<1 s) pulse of 467 nm light alters the eLOV protein conformation, unblocking the adjacent TEVcs and allowing cleavage. After ~100 s without blue light, eLOV spontaneously reverts to its dark state conformation and again blocks TEVcs.

To test the SPARK design with a well-established cellular PPI, we selected the β2-adrenergic receptor (β2AR), a 7-transmembrane GPCR that recruits the soluble cytosolic effector protein β-arrestin2 after stimulation with small-molecule agonist isoproterenol. We cloned β2AR and β-arrestin2 genes into the protein A and protein B positions of SPARK, respectively (*Figure 1B*). Note that β2AR is not fused to the tail domain of the vasopressin receptor or any other motif that artificially boosts its recruitment of arrestin (*Barnea et al., 2008*; *Lee et al., 2017*). HEK293T cells expressing these SPARK components were stimulated for 5 min with both 467 nm light (60 mW/cm$^2$) and isoproterenol. Under these conditions, we expect SPARK components to interact and release the transcription factor, Gal4, from the plasma membrane. Consequently, we should observe transcription and translation of the Citrine fluorescent protein reporter gene. Nine hours after stimulation of the HEK cells, we fixed them and imaged Citrine expression. *Figure 1C* shows robust Citrine expression in the light- and isoproterenol-treated cells. Essential for a functional 'AND' gate, we detected negligible Citrine expression in the light-only or isoproterenol-only cells.

Because SPARK is a transcriptional tool, the readout can be any gene of our choosing. Luciferase is frequently chosen for high-throughput assays, because it is easy to detect and quantify. By repeating the above experiment, but replacing the UAS-Citrine gene with a UAS-luciferase gene, and quantifying luminescence on a platereader, we again detected a robust increase in reporter gene

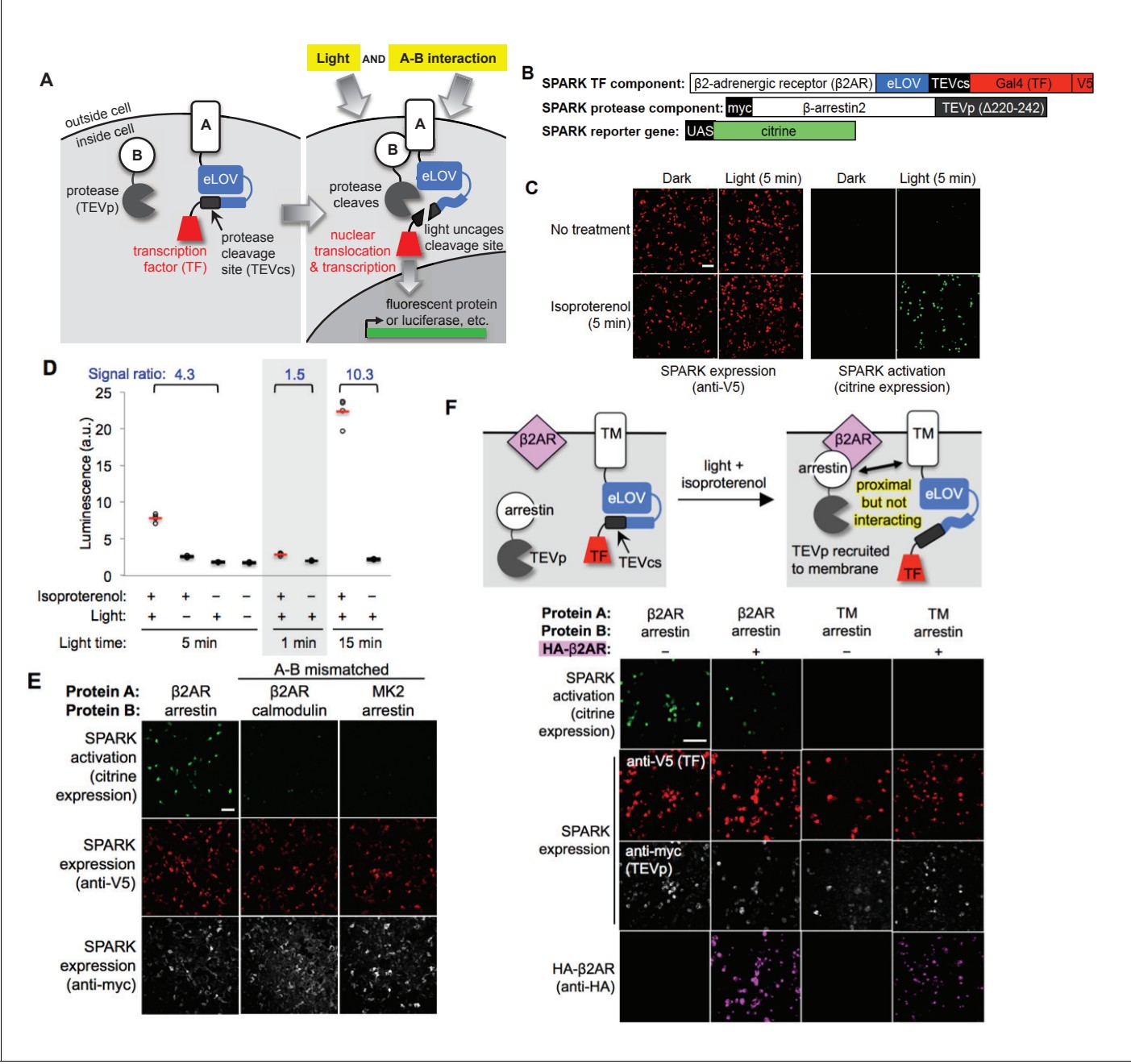

**Figure 1.** Design of SPARK and application to light- and agonist-dependent detection of β2-adrenergic receptor (β2AR)-β-arrestin2 interaction. (**A**) Scheme. A and B are proteins that interact under certain conditions. In this example, protein A is membrane-associated and is fused to a light-sensitive eLOV domain (*Wang et al., 2017*), a protease cleavage site (TEVcs), and a transcription factor (TF). These comprise the 'SPARK TF component.' Protein B is fused to a truncated variant of TEV protease (TEVp) ('SPARK protease component'). When A and B interact (right), TEVp is recruited to the vicinity of TEVcs. When blue light is applied to the cells, eLOV reversibly unblocks TEVcs. Hence, the coincidence of light *and* A-B interaction permits cleavage of TEVcs by TEVp, resulting in the release of the TF, which translocates to the nucleus and drives transcription of a chosen reporter gene. (**B**) SPARK constructs for studying the β2AR-β-arrestin2 interaction. V5 and myc are epitope tags. UAS is a promoter recognized by the TF Gal4. (**C**) Imaging of SPARK activation by β2AR-β-arrestin2 interaction under four conditions. HEK 293T cells were transiently transfected with the three SPARK components shown in (**B**). β2AR-β-arrestin2 interaction was induced with addition of 10 μM isoproterenol for 5 min. Light stimulation was via 467 nm LED at 60 mW/cm$^2$ and 10% duty cycle (0.5 s of light every 5 s) for 5 min. Nine hours after stimulation, cells were fixed and imaged. (**D**) Same as (**C**), but HEK 293T cells were stably expressing the SPARK protease component and transiently expressing SPARK TF component and UAS-luciferase. Results of shorter and longer irradiation times are also shown. ±isoproterenol signal ratio was quantified for each time point. Each datapoint reflects one well of a 96-well plate containing >6000 transfected cells. Four replicates per condition. (**E**) SPARK is specific for PPIs over non-interacting protein pairs. Same experiment as in (**C**), except arrestin was replaced by calmodulin protein (which does not interact with β2AR) in the second column, and β2AR was

*Figure 1 continued on next page*

*Figure 1 continued*

replaced by the calmodulin effector peptide MK2 (which does not interact with arrestin) in the third column. Anti-myc and anti-V5 antibodies stain for the SPARK protease and TF components, respectively. (F) SPARK is activated by direct interactions and not merely proximity. Top: experimental scheme. To drive proximity but not interaction, we created SPARK constructs in which A and B domains were a transmembrane (TM) segment of the CD4 protein, and β-arrestin2, respectively. TM and arrestin do not interact. HEK 293T cells expressing these SPARK constructs were also transfected with an expression plasmid for HA-tagged β2AR. Upon isoproterenol addition, β-arrestin2-TEVp is recruited to the plasma membrane via interaction with β2AR, but it does not interact directly with the SPARK TF component. Bottom: Images of HEK 293T cells 9 hr after stimulation with isoproterenol and light (for 5 min). The last column shows the experiment depicted in the scheme. The first two columns are positive controls with SPARK constructs containing β2AR and β-arrestin2 (which do interact). The third column is a negative control with omission of the HA-β2AR construct. Anti-V5, anti-myc, and anti-HA antibodies stain for SPARK TF component, SPARK protease component, and HA- β2AR proteins, respectively. All scale bars, 100 μm.

DOI: https://doi.org/10.7554/eLife.30233.002

The following figure supplements are available for figure 1:

**Figure supplement 1.** Characterization of SPARK – Testing alternative TEVcs sequences and alternative LOV domains.
DOI: https://doi.org/10.7554/eLife.30233.003

**Figure supplement 2.** Further characterization of SPARK tool.
DOI: https://doi.org/10.7554/eLife.30233.004

transcription in light- and isoproterenol-treated cells compared to light-only or isoproterenol-only cells (*Figure 1D*).

In an effort to further optimize SPARK, we explored alternative TEVcs sequences and alternative LOV domain sequences. Alternative TEVcs sequences, particularly with changes at the P1′ position immediately following the cleavage site, have the potential to alter both $k_{cat}$ and $K_m$ of the TEVp-TEVcs interaction. For optimal SPARK performance, we favor high $k_{cat}$, for high signal, and high $K_m$, for low background and strong proximity-dependence of the cleavage. Hence, we explored TEVcs variants with P1′=M, Q or Y because these have previously been shown to tune TEVp-TEVcs kinetics (*Wang et al., 2017*; *Kapust et al., 2002*). *Figure 1—figure supplement 1A* shows that P1′=M displayed the highest signal and strongest proximity dependence in the context of β2AR-β-arrestin2 SPARK, and hence we selected this TEVcs sequence for all subsequent experiments in this study.

While our evolved eLOV provides robust light caging of TEVcs (*Wang et al., 2017*), a new LOV sequence with improved properties over the original LOV (which served as the template for our directed evolution [*Wang et al., 2017*]) has been reported (*Lee et al., 2017*; *Guntas et al., 2015*). We thus performed a side-by-side comparison of this new LOV domain (called 'iLID') to our eLOV, in the context of β2AR-β-arrestin2 SPARK. In addition, we constructed two different hybrid LOV domains (hLOV1 and hLOV2) that combine features of eLOV and iLID into a single gene (*Figure 1— figure supplement 1B*). *Figure 1—figure supplement 1C and E* shows that with short stimulation times, or low SPARK component expression levels, eLOV is the best light gate, because it gives the highest signal in the +light condition. When SPARK components are highly overexpressed, and the stimulation time is increased from 5 min to 15 min, however, hLOV1 is the best light gate, because it gives the lowest signal in the no-light condition (i.e., best steric protection of TEVcs in the dark state, *Figure 1—figure supplement 1D*). Because we want SPARK to be robust even with short tagging time windows, we selected eLOV for all subsequent experiments in this study. For applications that require minimization of dark state leak, however, hLOV1 may prove to be a better light gate.

## Further characterization of SPARK in mammalian cells

We compared different light stimulation times and found that 5 min is sufficient to give a 4.3-fold increase in SPARK-driven luciferase expression (*Figure 1D*). 1 min of light is too short, and one can optionally increase the stimulation time to 15 min to further boost the ±isoproterenol signal ratio to 10.3. To test SPARK-PPI specificity, we replaced one protein partner in the β2AR-β-arrestin2 SPARK pair with a non-interacting protein. *Figure 1E* shows that the mismatched pairs β2AR-calmodulin and MK2-β-arrestin2 fail to drive Citrine expression.

We wondered whether mere recruitment of TEVp to the vicinity of TEVcs, without direct complexation mediated by the A-B PPI, would be sufficient to give proteolytic cleavage and TF release. To test this, we generated a mismatched SPARK pair using β-arrestin2 (protein B) and the transmembrane domain of CD4 (protein A). We transfected HEK cells with these SPARK constructs along with a V5-tagged β2AR expression plasmid. We then drove arrestin-TEVp translocation from the cytosol

to the plasma membrane by addition of isoproterenol (*Figure 1—figure supplement 2A*). *Figure 1F* and *Figure 1—figure supplement 2B* show that, even when arrestin-TEVp is recruited to the immediate vicinity of the TM (CD4) SPARK TF component, no SPARK activation (citrine expression) is observed. This demonstrates that SPARK is a very specific detector of direct A-B *interaction*, and is not activated merely by A-B *proximity*.

How efficient is SPARK? How much TEVcs cleavage occurs after our standard stimulation time of 5 min? To quantify this, we used the β2AR-β-arrestin2 SPARK constructs in *Figure 1B*, and lysed the cells immediately after stimulation to examine the SPARK proteins by Western blot. *Figure 1—figure supplement 2C* shows that 30% of total TEVcs is cleaved after 5 min of light and isoproterenol, and the cleavage yield increases to 48% after 30 min of stimulation.

For most of our experiments, we used a blue LED array to deliver light to cells. However, many laboratories do not have access to such a light source. The light power required to uncage the LOV domain is very weak, <0.5 mW/cm$^2$ (*Pudasaini and Zoltowski, 2013*). Thus we tested if SPARK could be stimulated equipotently by ordinary room light. *Figure 1—figure supplement 2D* shows that the signal ratios obtained are the same whether the light source is a blue LED or ambient room light. In some experiments in our study, we also used a broad-wavelength 'daylight lamp' to deliver light to SPARK-expressing cells. When we did not wish for light exposure, we kept the cells covered in aluminum foil and worked in a dark room with a red light source.

## SPARK can be generalized to a variety of PPIs

To test the generality of SPARK, we replaced β2AR and arrestin with a variety of other PPI pairs (*Figure 2A*). These include two different GPCRs - dopamine receptor D1 (DRD1) and neuromedin B receptor (NMBR) - that also recruit arrestin; a tyrosine kinase receptor that recruits Grb2 protein upon stimulation with a growth factor; the FKBP-FRB protein pair whose interaction is induced by rapamycin; and the light-regulated CRY2-CIBN protein pair. In contrast to β2AR, the latter four proteins are all soluble proteins that cannot by themselves sequester the TF component of SPARK in the cytosol. To ensure that the TF remains in the cytosol in the basal state, we fused transmembrane domains to both FKBP and CIBN.

*Figure 2B–C* and *Figure 2—figure supplement 1A* show that all protein pairs tested gave clear light- and ligand-dependent gene expression. Some background signal was observed in the no-light/+rapamycin condition for FRB/FKBP, but this is to be expected given that rapamycin dissociates very slowly (t$_{1/2}$ ~17.5 hours (*Hosoi et al., 1999*)) and consequently, the TEVp/TEVcs domains of these SPARK constructs remain in proximity for the entire 9 hr transcription/translation window. To test whether the TF component of SPARK would also work in a different subcellular region besides the plasma membrane, we created a FRB-SPARK construct localized to the outer mitochondrial membrane (OMM) (via fusion to the N-terminal 53-amino acid OMM targeting domain of AKAP1). *Figure 2B* shows that this SPARK pair gives rapamycin- and light-dependent gene expression as well.

GPCRs are a protein class of special interest due to their central role in signal transduction, their rich pharmacology, and their prevalence as therapeutic targets. Though ligand-activated GPCRs differ in their downstream recruitment of various G proteins (G$_i$, G$_s$, or G$_q$), nearly all of them recruit the effector arrestin as part of their desensitization pathway (*Shenoy and Lefkowitz, 2011*). As a consequence, assays that read out arrestin recruitment to GPCRs can detect activation of a wide range of GPCRs. We were interested to know whether SPARK could detect the ligand-dependent recruitment of arrestin to a range of GPCRs, and if so, whether the signal ratios would exceed those observed in the commonly-used GPCR assays TANGO (*Barnea et al., 2008*; *Kroeze et al., 2015*; *Zhou et al., 2017*) and DiscoverX (*Takakura et al., 2012*; *Southern et al., 2013*) (these typically give signal ratios of 2–3 or less). In *Figure 2D*, we selected eight different GCPRs and tested them in SPARK. The first six GPCRs gave robust light- and ligand-dependent luciferase expression, five of them with ±ligand signal ratios >15. Such large signal ratios are especially encouraging given that previous work has shown that some of these GPCRs (e.g., DRD2, NMBR and AVPR2) have significant interactions with arrestin prior to ligand addition (*Kroeze et al., 2015*), suggesting that background signal (SPARK-driven luciferase expression in the no ligand condition) could have been a concern. Despite this, the high sensitivity and specificity of SPARK permitted facile differentiation between the +ligand and no-ligand states.

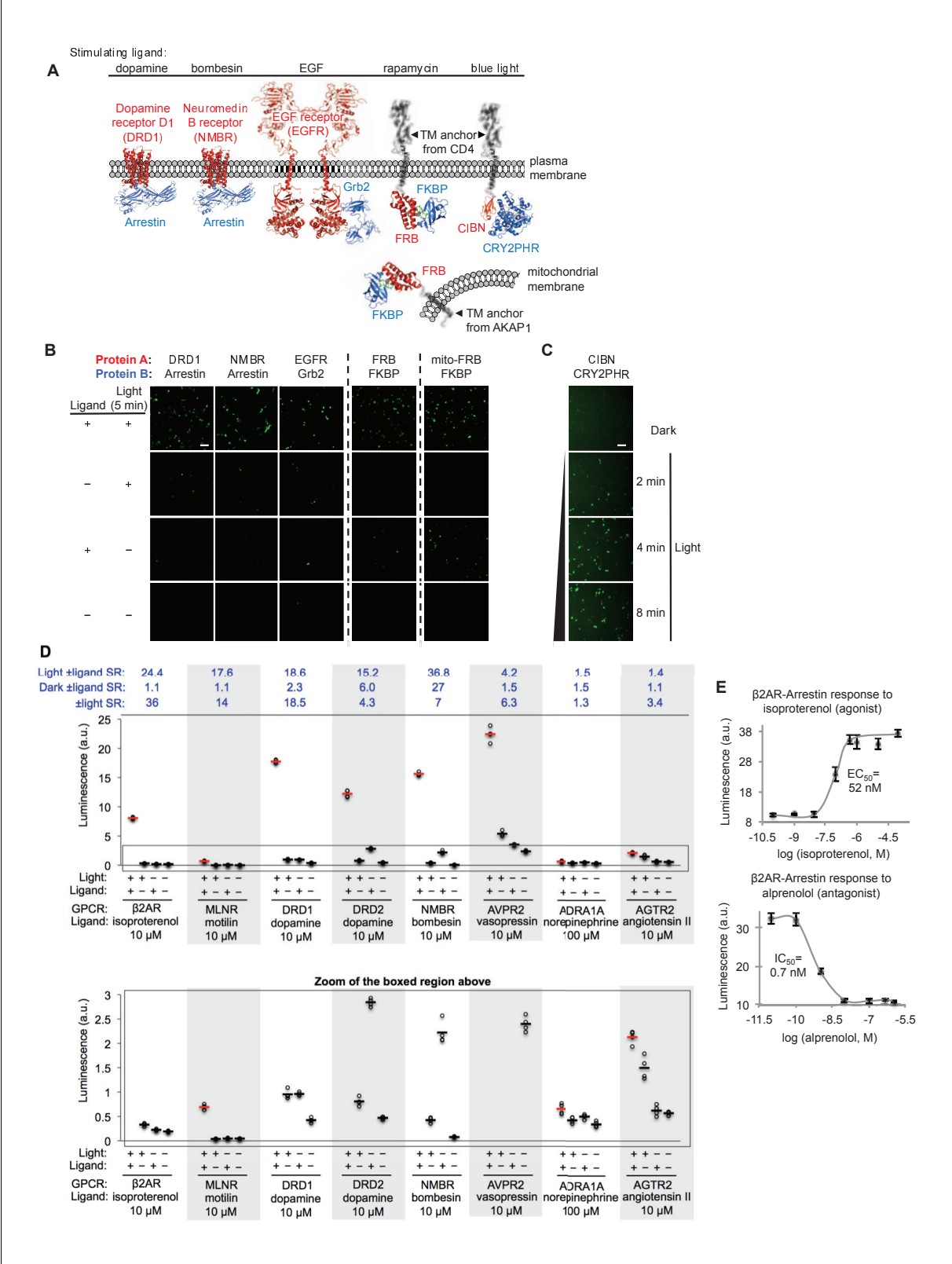

**Figure 2.** SPARK can be applied to a variety of PPIs. (**A**) PPI pairs studied with SPARK. DRD1 and NMBR are GPCRs that interact with β-arrestin2. EGFR is a receptor tyrosine kinase that recruits Grb2 upon stimulation with EGF ligand. FKBP and FRB are soluble proteins that heterodimerize upon addition of the drug rapamycin; to keep FRB SPARK out of the nucleus in the basal state, we fuse it to either a plasma membrane anchor (TM from CD4) or a mitochondrial membrane anchor (TM from AKAP1). CIBN-CRY2PHR is a light-inducible PPI (***Kennedy et al., 2010***). (**B**) SPARK data corresponding to

*Figure 2 continued on next page*

*Figure 2 continued*

PPIs depicted in (**A**). SPARK constructs were the same as those shown in *Figure 1B*, except β2AR and β-arrestin2 genes were respectively replaced by the A and B protein-coding genes indicated. HEK293T cells transiently expressing SPARK constructs were stimulated with light and the ligand indicated in (**A**) for 5 min, then fixed and imaged 9 hr later. Citrine fluorescence images are shown. Dashed lines separate experiments that were performed separately and shown with different Citrine intensity scales. Scale bar, 100 μm. Quantitation of EGFR-Grb2 SPARK images (five separate fields of view): ±light signal ratio = 12; ±EGF signal ratio = 27. (**C**) SPARK detection of CIBN-CRY2PHR interaction. Blue light (467 nm, 60 mW/cm², 33% duty cycle (2 s light every 6 s)) simultaneously uncages the eLOV domain and induces the CIBN-CRY2PHR interaction. Scale bar, 100 μm. See *Figure 2—figure supplement 1* for additional data using FRB-FKBP and CIBN-CRY2PHR SPARK constructs. (**D**) SPARK applied to eight different GPCRs. HEK293T cells were prepared as in *Figure 1D*. The SPARK protease component is β-arrestin2-TEVp. The SPARK TF component contains the indicated GPCR (no vasopressin V2 domain). Light (ambient) and ligand were applied for 15 min total, then cells were analyzed for luciferase activity 9 hr later. Four replicates per condition. ±Ligand signal ratios (SR) and ±light signal ratios for each GPCR quantified across top. The boxed region is enlarged at bottom. (**E**) Isoproterenol and alprenolol dose-response curves with β2AR-β-arrestin2 SPARK readout. HEK293T cells were prepared and stimulated as in *Figure 1D*, with 5 min light window. Four replicates per concentration. Errors, ±STD. $EC_{50}$ of 52 nM for isoproterenol and $IC_{50}$ of 0.7 nM for alprenolol are close to published values (*Fisher et al., 2010*; *Gether et al., 1995*).

DOI: https://doi.org/10.7554/eLife.30233.005

The following figure supplement is available for figure 2:

**Figure supplement 1.** Additional data with FRB-FKBP and CIBN-CRY2PHR SPARK constructs.

DOI: https://doi.org/10.7554/eLife.30233.006

The last two GPCRs we surveyed, AGTR2 and ADRA1A, are not known to recruit arrestin, and previous assays (e.g., PRESTO-TANGO (*Kroeze et al., 2015*)) failed to produce signal. Our SPARK ±ligand signal ratios were small for these GPCRs (1.5 and 1.4, respectively, *Figure 2D*) but statistically significant (p<0.001 for both in two separate experiments). Thus the sensitivity of SPARK enables us to conclude that these receptors do indeed recruit arrestin, but to a lesser extent than other GPCRs.

In addition to detecting the presence of ligands, scientists studying GPCRs are often interested in differentiating agonists from antagonists, and calculating EC50 and IC50 values. In *Figure 2E*, we show that SPARK can be easily used to generate dose-response curves for the β2AR ligands isoproterenol (an agonist) and alprenolol (an antagonist).

In total, *Figure 2* extends SPARK to 12 different PPI pairs. Importantly, these results were all obtained simply by cloning protein-coding genes into positions A and B of the constructs in *Figure 1A*. We did not vary geometries or linkers or create panels of constructs to test for each PPI of interest. The success of all 12 PPI pairs on the first try demonstrates that the SPARK scaffold is robust and highly modular in its design.

## SPARK for probing PPI dynamics

Because SPARK is gated by light, and light can be applied during any user-specified time window, we envisioned staggering the light window to read out different temporal regimes of a PPI time course. For instance, if an A-B PPI has the time course shown in *Figure 3A*, left, a light window that overlaps with the peak of A-B interaction will result in SPARK activation and gene transcription, whereas a light window that is right-shifted to a later time period (*Figure 3A*, right) will not. To test this concept, we used the β2AR-arrestin2 SPARK constructs, applied isoproterenol at t = 0, and read out luciferase expression after shifting the light window to various time ranges (e.g., 0–5 min, 5–10 min, 15–20 min, etc.). *Figure 3C* shows that the β2AR-β-arrestin2 interaction extent is high in the initial time period (0–5 min), and then falls off rapidly in the 5–10 min range. This is consistent with previous reports showing that the β2AR-arrestin interaction peaks between 1–3 minutes and then falls off (*Reiner et al., 2010*; *Eichel et al., 2016*; *Lobingier et al., 2017*).

β2AR is a GPCR with multiple known ligands (*Figure 3B*). Isoproterenol and isoetharine are both full agonists, while clenbuterol is a partial agonist, evoking a weaker downstream response and kinetically slower recruitment of arrestin (*Kaya et al., 2012*). Carvedilol is an inverse agonist, which can attenuate downstream signaling while still recruiting arrestin, albeit more weakly than a full agonist (*Wisler et al., 2007*). We used the light staggering scheme in *Figure 3A* to examine the temporal response of the β2AR-arrestin2 interaction to these different ligands. As expected, the full agonists isoproterenol and isoetharine evoked strong responses (maximum ±ligand signal ratios ~18) and transient arrestin association. Clenbuterol induced a much smaller response and delayed onset

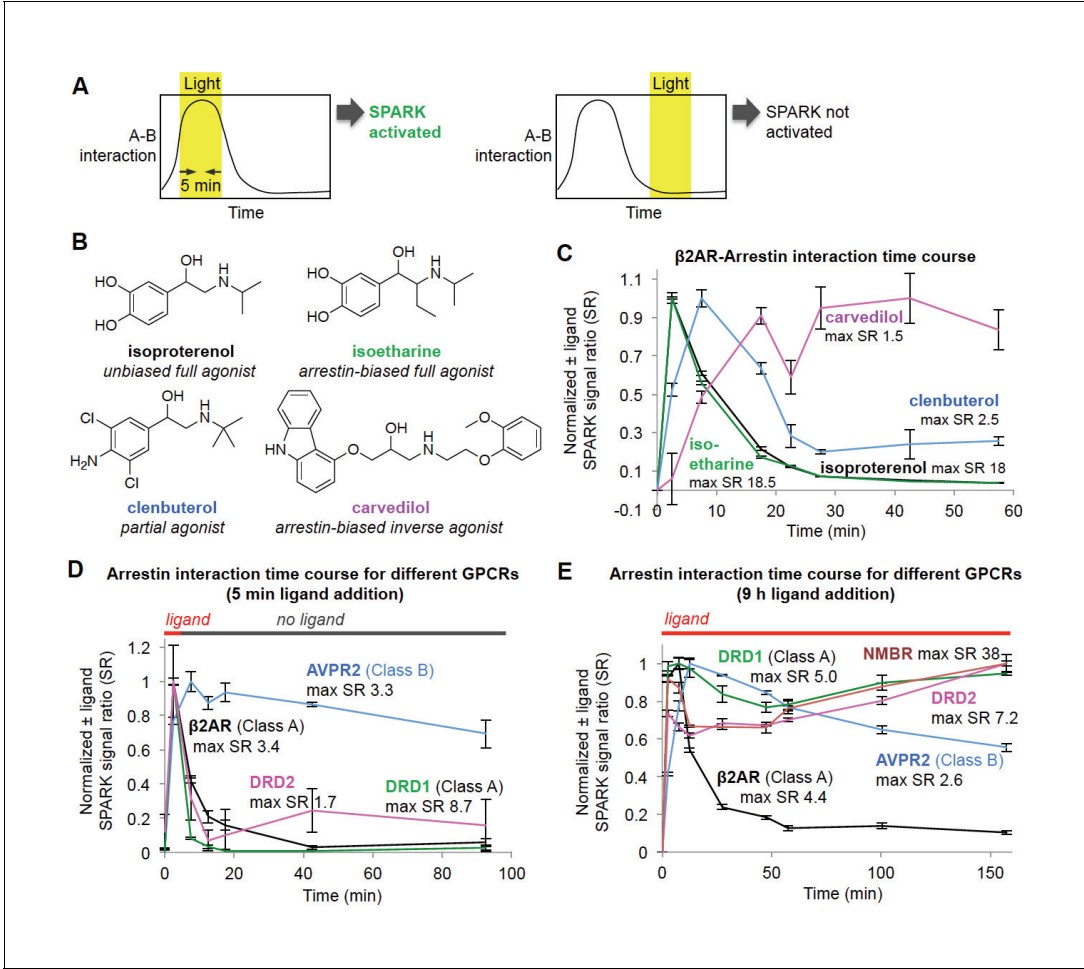

**Figure 3.** Shifting the light window permits SPARK analysis of the dynamic GPCR-β-arrestin2 interaction. (**A**) Scheme. By shifting the light window, it is possible to read out different temporal regimes of protein A-protein B interaction. On the left, light coincides with a period of high A-B interaction, resulting in SPARK activation and transcription of a reporter gene. On the right, light coincides with a period of low A-B interaction, so SPARK is not activated. (**B**) Panel of β2AR agonists, partial agonists, and antagonist. Biased agonists preferentially recruit one downstream effector (such as β-arrestin2) over another. (**C**) β2AR-β-arrestin2 interaction time course with various ligands. HEK293T cells expressing SPARK constructs were prepared as in *Figure 1D*. 15 hr after transfection, 10 µM ligand was added at time = 0 min and remained on the cells for the duration of the experiment. The light window was 5 min, centered around the timepoint given on the x axis. 9 hr after initial addition of ligand, cells were mixed with luciferin substrate and analyzed for luciferase activity. Time courses are normalized with maximum signal ratio (SR) set to one and minimum SR set to 0. Each datapoint represents the mean of 4 replicates. Errors, ±STD. (**D**) Receptor-β-arrestin2 interaction time course with different receptors. Same as (**C**), except β2AR in the SPARK TF component was replaced with various other GPCRs, and ligand was added only briefly, from time = 0 to 5 min. (**E**) Same as (**D**), except ligand remained on the cells for the duration of the experiment (9 hr).

DOI: https://doi.org/10.7554/eLife.30233.007

of arrestin recruitment, consistent with its classification as a partial agonist. Interestingly, carvedilol produced a delayed but sustained interaction with β-arrestin2.

Different GPCRs are also thought to recruit arrestin with different kinetics. We performed the same SPARK time course assay, keeping the β-arrestin2-TEVp SPARK component constant, but varying the identity of the GPCR. *Figure 3D* shows that two Class A GPCRs, β2AR and DRD1, give strong SPARK responses within 5 min of ligand addition but much lower SPARK-driven luciferase expression shortly after ligand washout. This is consistent with previous studies showing that Class A GPCRs interact weakly and transiently with arrestin (*Oakley et al., 2000*). In contrast, the Class B GPCR AVPR2 recruits arrestin quickly and maintains a strong interaction with arrestin even 90 min after washout of the ligand vasopressin. The dopamine receptor DRD2 has not previously been classified as type A or B, but our data in *Figure 3D* strongly suggest that it is Class A, like DRD1, due to

its transient association with arrestin. *Figure 3E* shows the same experiment, but with ligand left on the cells for the entire duration of the experiment. Now, SPARK-driven luciferase expression is observed even at 150 min for DRD1 and DRD2, but this is likely due to continual stimulation of fresh surface receptor pools with dopamine.

We conclude that, due to its time-gated transcriptional design, SPARK can resolve PPI dynamics, with a temporal resolution of 5 min, via a single timepoint readout that is simple and scalable (our assays in *Figure 3* were performed in 96-well plate format). Excitingly, this feature enabled us to classify a receptor (DRD2) important in emotion, cognition, and neurological disease as a Class A-type GPCR.

## SPARK for genetic selections

A powerful application of transcriptional assays such as yeast two-hybrid (*Fields and Song, 1989*; *Luo et al., 1997*) is to perform cell-based selections for PPI discovery. In these experiments, each cell in a population expresses a constant 'bait' protein and a variable 'prey' protein. If in a particular cell, the prey interacts with bait, a transcription factor is reconstituted, and transcription of a survival gene such as HIS3 ensues. However, existing two-hybrid assays have two major limitations: high background, resulting in many false positive identifications, and no temporal resolution: is it impossible to distinguish between PPIs that occurred 1 min after ligand addition versus 20 min after ligand addition, for example. These problems have prevented the application of existing PPI-dependent transcriptional assays, including split-ubiquitin (*Petschnigg et al., 2014*; *Fetchko and Stagljar, 2004*) and TANGO (*Barnea et al., 2008*; *Inagaki et al., 2012*), to genetic selections in mammalian cells. SPARK offers a potential solution, because the light gate both reduces background/enhances specificity, *and* provides temporal precision, allowing the user to selectively focus on PPIs that occur during a 10 min time window of their choosing.

We devised a preliminary 'model selection' to see if SPARK had sufficiently high dynamic range, in combination with Fluorescence Activated Cell Sorting (FACS), to selectively enrich for cells expressing a matched PPI pair versus a mismatched, non-interacting protein pair (*Figure 4*). We prepared HEK cells expressing the β2AR-β-arrestin2 SPARK components in *Figure 4A* (matched PPI), and separate HEK cells expressing a mismatched β2AR-calmodulin SPARK pair, and mixed the two cell populations together in a 1:50 ratio (*Figure 4A*). We then stimulated with isoproterenol and light for 10 min, and cultured for nine additional hours to allow transcription and translation of the reporter gene, Citrine. Then cells were lifted and sorted on a FACS instrument to enrich cells with the highest Citrine expression. We analyzed the resulting cell population by RT-qPCR to determine the ratio of arrestin cells (matched SPARK) to calmodulin cells (mismatched SPARK). *Figure 4C* shows that a single round of FACS sorting enriched the matched PPI cell population by 51-fold over the mismatched cell population. Such a large enrichment factor in a single round of selection suggests that it should be possible to use SPARK to accurately discover PPIs that occur during selected time windows of interest.

## Comparison of SPARK to TANGO and iTANGO

TANGO is a transcriptional assay for reading out receptor activation that, like SPARK, uses PPI-driven proteolytic release of a transcription factor (*Barnea et al., 2008*; *Kroeze et al., 2015*). In contrast to SPARK, however, TANGO lacks a light gate, and it utilizes a higher affinity TEVp-TEVcs pair (with $K_m$ of 240 µM versus est. 450 µM for our system [*Kapust et al., 2002*; *Kapust et al., 2001*]) that reduces the dynamic range and increases the background of the assay. Furthermore, when applied to GPCRs, TANGO requires a vasopressin receptor 2 tail domain (V2 tail) to be fused to the GPCR's C-terminal end, to artificially enhance arrestin recruitment and increase TANGO signal. Because TANGO is used extensively in the GPCR field for its simple and scalable design (*Kroeze et al., 2015*), we performed a side-by-side comparison to SPARK. β2AR-arrestin2 SPARK and TANGO constructs used for the comparison are shown in *Figure 4—figure supplement 1A*.

HEK293T cells expressing the respective constructs were treated with light and isoproterenol for 15 min, then cultured for 9 hr, before quantitation of luciferase expression (*Figure 4—figure supplement 1B*). The ±ligand signal ratio was 16.4-fold higher for SPARK than for TANGO under these conditions. Since the TANGO assay is typically performed with much longer periods of ligand stimulation (*Barnea et al., 2008*; *Kroeze et al., 2015*), we repeated the assay but increased the

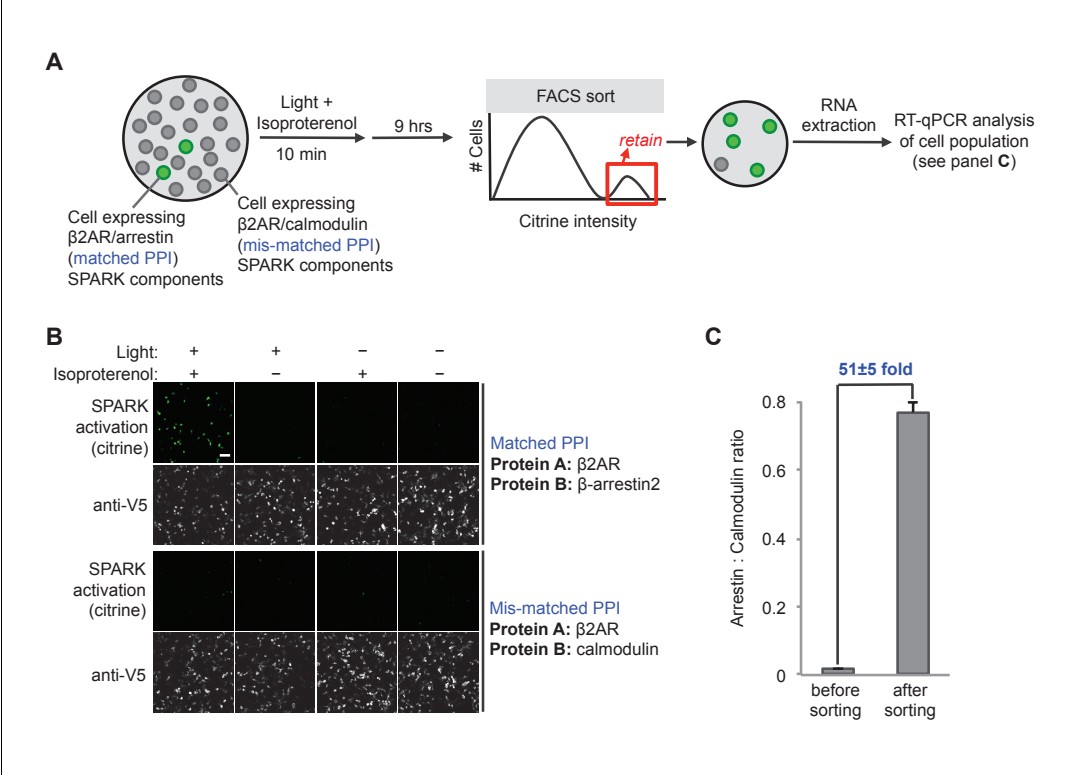

**Figure 4.** SPARK can be coupled to genetic selections. (**A**) Selection scheme to enrich cells with PPI (protein-protein interaction) event during time window of interest over cells without PPI event. Culture dish containing mixed population of cells. Protein A (the 'bait') remains constant in the SPARK TF component, but protein B ('prey') varies in the SPARK protease component. In our model selection, we used β-arrestin2 as the matched prey (interacts with β2AR), and calmodulin as the mis-matched prey (does not interact with β2AR). The culture dish is subjected to light and β2AR-stimulating ligand isoproterenol for 10 min, then cultured for an additional 9 hr. FACS sorting is performed to isolate cells with high Citrine expression, indicative of SPARK activation. After sorting, the cell population is expected to be strongly enriched in β-arrestin2-expressing cells over calmodulin-expressing cells. (**B**) Imaging of HEK293T cells expressing SPARK components with matched bait + prey (top) versus mismatched bait + prey (bottom). Images were acquired 9 hr after stimulation with 10 μM isoproterenol and light (daylight lamp, 25W, 6500K, 480 nm/530 nm/590 nm). Anti-V5 antibody stains for the SPARK β2AR TF component. Scale bar, 100 μm. (**C**) RT-qPCR analysis of mixed cell population before and after FACS sorting. RNA was extracted from cells and reverse transcribed, and arrestin-TEVp:calmodulin-TEVp ratios were quantified by qPCR. Pre-sort ratio is 1:66; post-sort ratio is 1:1.3. Data represent the mean of three replicates. Error bar, ±STD.

DOI: https://doi.org/10.7554/eLife.30233.008

The following figure supplement is available for figure 4:

**Figure supplement 1.** SPARK comparison to TANGO and iTANGO.

DOI: https://doi.org/10.7554/eLife.30233.009

isoproterenol incubation to 18 hr (light period was still 15 min for SPARK). The TANGO ± ligand signal ratio increased from 1.2 to 1.5, but was still far lower than the SPARK signal ratio, which was 11.8. The main reason for this difference in performance appears to be high TANGO background in the no-ligand condition. SPARK has much lower no-ligand luciferase expression because the eLOV domain protects the TEVcs from cleavage in the dark state. In contrast, TANGO's TEVcs is exposed throughout the entire period (hours to days) that the TANGO components are expressed, leading to high accumulated background.

Recently, a light-gated version of TANGO was reported, called iTANGO (*Lee et al., 2017*). iTANGO was used to mark neurons exposed to dopamine in the mouse brain (*Lee et al., 2017*). Though not explored for PPI detection, the design of iTANGO suggests that it could potentially also be applied to PPI detection. Thus, we performed a side-by-side comparison of iTANGO and SPARK as well, for the detection of isoproterenol-stimulated β2AR-arrestin interaction. The constructs used for the comparison are shown in *Figure 4—figure supplement 1A*, and the results are shown in *Figure 4—figure supplement 1C*. We found that under all conditions tested, iTANGO gave

significantly higher background than SPARK. For example, there was 3 to 26-fold greater iTANGO-driven luciferase expression under dark/+ligand and light/no-ligand conditions, than SPARK-driven luciferase expression under the same conditions. This resulted in 2 to 3-fold greater ±ligand signal ratios for SPARK, and 5 to 14-fold greater ±light signal ratios, compared to iTANGO (*Figure 4—figure supplement 1C*).

We hypothesize that the high background we observed with iTANGO derives mainly from its use of split TEV protease, whose fragments may have high affinity for one another, and reverse slowly, similar to other split proteins (*Martell et al., 2016*; *Kerppola, 2008*). To test this possibility, we performed GPCR-arrestin time course assays using both SPARK and iTANGO. *Figure 4—figure supplement 1D and E* show that SPARK signal reverses quickly after ligand washout, whereas iTANGO continues to gives signal for 15–45 min longer. This is consistent with iTANGO's design 'trapping' PPIs, while SPARK's design is much less perturbative.

Based on our collective observations, we conclude that SPARK is more sensitive, specific, and accurate than TANGO and iTANGO for PPI detection in cells.

## Discussion

The SPARK assay we have developed converts a specific, transient molecular interaction into a stable and amplifiable reporter gene signal. As such, SPARK is highly versatile, able to drive luciferase expression for platereader assays, or fluorescent protein expression for microscopy or FACS-based cell selections. Though not shown here, it should be facile to couple SPARK to the expression of other reporter genes as well, including APEX for EM (*Martell et al., 2012*; *Lam et al., 2015*) and proteomics (*Rhee et al., 2013*), antibiotic resistance genes, and various actuators such as opsins and toxins.

To develop SPARK, we optimized both the protease cleavage sequence and the light-sensitive LOV domain. The resulting tool is highly modular and generalizable; 12 of the 13 PPIs (including transmembrane as well as soluble proteins) we cloned into SPARK worked on the first try, without any optimization of geometries or linkers. SPARK is highly specific for direct PPIs, and is not activated merely by protein proximity (e.g., colocalization to the plasma membrane – *Figure 1F*). We showed that only 5 min of light coinciding with a PPI event are sufficient to produce a stable signal that lasts for nine hours to days.

SPARK produces much larger signal-to-noise ratios than other PPI assays. When Presto-TANGO was applied to 167 different GPCRs, for example, the majority of them gave signal ratios between 1.3 and 5 (Ref. *Kroeze et al., 2015*). In contrast, SPARK gave signal ratios >15 for five of the eight GPCRs that we randomly selected, and signal ratios >10 for other PPI pairs (*Figure 2B and C*). The higher signal ratios result from two factors. First, noise is lower for SPARK because the LOV domain prevents proteolytic cleavage outside of the brief light window. Second, signal is higher for SPARK because our TEVp-TEVcs pair has higher catalytic turnover (est. 10.8 $min^{-1}$ for SPARK compared to 0.84 $min^{-1}$ for TANGO (Kapust et al., 2002); and iTANGO uses split-TEVp (*Lee et al., 2017*) which is expected to have even slower turnover). As a result of improved signal ratio, SPARK can be used without artificial enhancement of native affinity (e.g., V2 tail), in contrast to both TANGO and iTANGO. And due to its higher sensitivity, SPARK can detect the activation of receptors that are opaque to other methods, such as ADRA1A and AGTR2 in *Figure 2D*, which gave no ±ligand signal difference with Presto-TANGO (*Kroeze et al., 2015*).

FRET is also commonly used to image cellular PPIs, and has been applied to GPCR-arrestin interactions in particular (*Nuber et al., 2016*). SPARK also gives much larger signal-to-noise ratios than those observed in such FRET experiments (1.02–1.03 S/N in *Nuber et al., 2016*), partly because SPARK incorporates signal amplification. SPARK is also more accessible to most laboratories, as it does not require specialized instrumentation or software as FRET imaging does.

Protein Complementation Assays (PCAs) are another commonly used alternative assay to detect and study cellular PPIs. Irreversible PCAs such as bimolecular fluorescence complementation (BiFC) (*Kerppola, 2008*) and split HRP (*Martell et al., 2016*) also integrate molecular events and produce stable signals, but unlike SPARK, they trap PPI partners in an irreversible complex that can perturb cellular signaling and give rise to false positives. Reversible PCAs such as split luciferase (*Ozawa et al., 2001*) do not trap, but they also do not produce stable signals that enable single

time point readout or genetic selections as SPARK does. By integrating through a transcription factor, SPARK is unique in combining a non-trapping design with stable, integrated signal generation.

PPI dynamics are central to signal transduction but previous transcriptional assays were unable to probe this aspect of their biology. Due to SPARK's light gate, we showed that it is possible to 'time stamp' PPI events, and even to reconstruct PPI time courses.

Future applications of SPARK might include examination of GPCR 'biased' signaling (*Bologna et al., 2017*). Here, we used only arrestin-TEVp with our GPCR panel, but it should be possible to replace the arrestin gene with Gi, Gs, or Gq and compare intensities and time courses of G protein versus arrestin recruitment to GPCRs in response to various ligands. Our model selection using FACS also suggests that SPARK will have utility for PPI discovery and other functional genomics applications. Notably, while yeast two hybrid and split ubiquitin assays have been employed for yeast-based selections (*Fetchko and Stagljar, 2004*; *Ito et al., 2001*), we are not aware of any applications of these platforms or related ones to mammalian cell-based selections, perhaps because the signal-to-noise ratios of these assays are not sufficiently high to enable selection in mammalian cells.

## Materials and methods

### Plasmids and cloning

All constructs for transient expression were cloned into pAAV vector. For stable expression, the constructs were cloned into pLX208. *Table 1* lists all 35 plasmids used in this work.

Rat β-arrestin2 was amplified from rat β-arrestin2-EGFP (Addgene plasmid #14697, Robert Lefkowitz laboratory). β2AR was amplified from Flag-β2AR (Addgene plasmid #14697, Robert Lefkowitz laboratory). The EGFR gene was amplified from Addgene plasmid #11011, Matthew Meyerson laboratory. Grb2 was amplified from pGEX-Grb2 (Addgene plasmid #46442, Bruce Mayer laboratory). Luciferase was amplified from GAL4UAS-Luciferase reporter (Addgene plasmid #64125, Moritoshi Sato). DRD1, DRD2, NMBR, MLNR, AVPR2, AGTR2, ADRA1A genes were each amplified from DRD1-TANGO, DRD2-TANGO, NMBR-TANGO, MLNR-TANGO, AVPR2-TANGO, AGTR2-TANGO, and ADRA1A-TANGO, respectively (Addgene plasmid #s 66268, 66269, 66445, 66434, 66227, 66223, 66213, Bryan Roth laboratory). hLOV1 and hLOV2 genes were synthesized by IDT.

Standard cloning procedures were used. PCR fragments were amplified using Q5 polymerase (NEB). The vectors were double-digested with NEB restriction enzymes and ligated to gel-purified PCR products by T4 ligation or Gibson assembly. Ligated plasmid products were introduced by heat shock transformation into competent XL1-Blue bacteria.

### HEK293T cell culture

HEK293T cells from ATCC with fewer than 20 passages were cultured as monolayers in complete growth media: DMEM (Dublecco's Modified Eagle medium, Gibco) supplemented with 10% FBS (Fetal Bovine Serum, Sigma) and 1% (v/v) Penicillin-Streptomycin (VWR, 5000 units/ml of penicillin and 5000 µg/mL streptomycin), at 37°C under 5% $CO_2$. For imaging at 10x magnification, we grew the cells in plastic 48-well plates that were pretreated with 50 µg/mL human fibronectin (Millipore) for at least 10 min at 37°C before cell plating (to improve adherence of HEK cells). For imaging at 40x magnification, we grew cells on 7 × 7 mm glass coverslips placed inside 48-well plates. The coverslips were also pretreated with 50 µg/mL human fibronectin for at least 10 min at 37°C before cell plating. Cells were plated at such a density that they would reach 60–90% confluence the next day.

### Fluorescence microscopy of cultured cells

Confocal imaging was performed on a Zeiss AxioObserver inverted confocal microscope with 10x air and 40x oil-immersion objectives, outfitted with a Yokogawa spinning disk confocal head, a Quad-band notch dichroic mirror (405/488/568/647), and 405 (diode), 491 (DPSS), 561 (DPSS) and 640 nm (diode) lasers (all 50 mW). The following combinations of laser excitation and emission filters were used for various fluorophores: Citrine/Alexa Fluor 488 (491 laser excitation; 528/38 emission), mCherry/Alexa Fluor 568 (561 laser excitation; 617/73 emission), Alexa Fluor 647 (647 excitation; 680/30 emission), and differential interference contrast (DIC). Acquisition times ranged from 100 to 500 ms. All images were collected and processed using SlideBook (Intelligent Imaging Innovations).

**Table 1.** Genetic constructs used in this study.

| Name | Features | Promoter/Vector | Details |
|------|----------|-----------------|---------|
| P1 | UAS-Citrine | UAS/pAAV | UAS promoter driving Citrine expression<br>SPARK reporter construct |
| P2 | UAS-Luciferase | UAS/pAAV | UAS promoter driving luciferase expression<br>SPARK reporter construct |
| P3 | Myc-Rat-β-arrestin2-TEVp Δ220–242 | CMV/pAAV | Myc: EQKLISEEDL<br>SPARK protease construct for transient expression with Myc epitope tag |
| P4 | Rat-β-arrestin2-HA-TEVp Δ220–242 | CMV/pLX208 | Hygromycin selection marker<br>HAx2: YPYDVPDYAYPYDVPDYA<br>SPARK protease construct for stable integration |
| P5 | Rat-β-arrestin2-HA-TEVp Δ220–242 | CMV/pAAV | HAx2: YPYDVPDYAYPYDVPDYA<br>SPARK protease construct for transient expression with HA epitope tag |
| P6 | β2AR-eLOV-TEVcs-FLAG-GAL4-V5 | CMV/pAAV | TEVcs: ENLYFQ**M**<br>FLAG: DYKDDDDK<br>V5: GKPIPNPLLGLDST<br>SPARK GPCR (β2AR) construct |
| P7 | Myc-CaM-TEVp Δ220–242 | CMV/pAAV | Myc: EQKLISEEDL<br>Used in SPARK specificity experiment (*Figure 1E*) |
| P8 | TM(CD4)-CIBN-MK2-eLOV-TEVcs-FLAG-GAL4-V5 | CMV/pAAV | TEVcs: ENLYFQ**M**<br>FLAG: DYKDDDDK<br>V5: GKPIPNPLLGLDST<br>Used in SPARK specificity experiment (*Figure 1E*) |
| P9 | Rat-β-arrestin2-EGFP | CMV/pEGFP | Used in arrestin translocation experiment (*Figure 1—figure supplement 2A*) |
| P10 | HA-β2AR | CMV/pAAV | Cleavable signal sequence of influenza hemagglutinin: MKTIIALSYIFCLVFA<br>HAx2: YPYDVPDYAYPYDVPDYA<br>Used in Figure 1F |
| P11 | β2AR-eLOV-TEVcs(Y)-FLAG-GAL4-V5 | CMV/pAAV | TEVcs(Y): ENLYFQ**Y**<br>SPARK GPCR (β2AR) construct with alternative TEVp cleavage site |
| P12 | β2AR-eLOV-TEVcs(Q)-FLAG-GAL4-V5 | CMV/pAAV | TEVcs(Q): ENLYFQ**Q**<br>SPARK GPCR (β2AR) construct with alternative TEVp cleavage site |
| P13 | ADRA1A-eLOV-TEVcs-FLAG-GAL4-V5 | CMV/pAAV | TEVcs: ENLYFQ**M**<br>FLAG: DYKDDDDK<br>V5: GKPIPNPLLGLDST<br>SPARK GPCR (ADRA1A) construct |
| P14 | AGTR2-eLOV-TEVcs-FLAG-GAL4-V5 | CMV/pAAV | TEVcs: ENLYFQ**M**<br>FLAG: DYKDDDDK<br>V5: GKPIPNPLLGLDST<br>SPARK GPCR (AGTR2) construct |
| P15 | AVPR2-eLOV-TEVcs-FLAG-GAL4-V5 | CMV/pAAV | TEVcs: ENLYFQ**M**<br>FLAG: DYKDDDDK<br>V5: GKPIPNPLLGLDST<br>SPARK GPCR (AVPR2) construct |
| P16 | DRD1-eLOV-TEVcs-FLAG-GAL4-V5 | CMV/pAAV | TEVcs: ENLYFQ**M**<br>FLAG: DYKDDDDK<br>V5: GKPIPNPLLGLDST<br>SPARK GPCR (DRD1) construct |
| P17 | DRD2-eLOV-TEVcs-FLAG-GAL4-V5 | CMV/pAAV | TEVcs: ENLYFQ**M**<br>FLAG: DYKDDDDK<br>V5: GKPIPNPLLGLDST<br>SPARK GPCR (DRD2) construct |
| P18 | EGFR-eLOV-TEVcs-FLAG-GAL4-V5 | CMV/pAAV | TEVcs: ENLYFQ**M**<br>FLAG: DYKDDDDK<br>V5: GKPIPNPLLGLDST<br>SPARK EGFR construct |
| P19 | NMBR-eLOV-TEVcs-FLAG-GAL4-V5 | CMV/pAAV | TEVcs: ENLYFQ**M**<br>FLAG: DYKDDDDK<br>V5: GKPIPNPLLGLDST<br>SPARK GPCR (NMBR) construct |

*Table 1 continued on next page*

Table 1 continued

| Name | Features | Promoter/Vector | Details |
|------|----------|-----------------|---------|
| P20 | MLNR-eLOV-TEVcs-FLAG-GAL4-V5 | CMV/pAAV | TEVcs: ENLYFQ**M**<br>FLAG: DYKDDDDK<br>V5: GKPIPNPLLGLDST<br>SPARK GPCR (MLNR) construct |
| P21 | β2AR-hLOV2-TEVcs-FLAG-GAL4-V5 | CMV/pAAV | TEVcs: ENLYFQ**M**<br>FLAG: DYKDDDDK<br>V5: GKPIPNPLLGLDST<br>SPARK GPCR (β2AR) construct with alternative LOV domain |
| P22 | β2AR-hLOV1-TEVcs-FLAG-GAL4-V5 | CMV/pAAV | TEVcs: ENLYFQ**M**<br>FLAG: DYKDDDDK<br>V5: GKPIPNPLLGLDST<br>SPARK GPCR (β2AR) construct with alternative LOV domain |
| P23 | β2AR-iLID$_M$-TEVcs(M)-FLAG-GAL4-V5 | CMV/pAAV | TEVcs(M): ENLYFQ**M**<br>FLAG: DYKDDDDK<br>V5: GKPIPNPLLGLDST<br>SPARK GPCR (β2AR) construct with alternative LOV domain |
| P24 | β2AR-iLID$_G$-TEVcs(G)-FLAG-GAL4-V5 | CMV/pAAV | TEVcs(G): ENLYFQ**G**<br>FLAG: DYKDDDDK<br>V5: GKPIPNPLLGLDST<br>SPARK GPCR (β2AR) construct with alternative LOV domain |
| P25 | Calmodulin-HA-TEVp Δ220–242 | CMV/pLX208 | Hygromycin selection marker<br>HAx2: YPYDVPDYAYPYDVPDYA<br>SPARK protease construct for stable integration |
| P26 | β2AR-V2-eLOV-TEVcs-FLAG-GAL4-V5 | CMV/pAAV | TEVcs: ENLYFQ**M**<br>FLAG: DYKDDDDK<br>V5: GKPIPNPLLGLDST<br>V2: GRTPPSLGPQDESCTTASSSLAKDTSS<br>Used in *Figure 4—figure supplement 1* |
| P27 | β2AR-TEVcs(L)-FLAG-GAL4-V5 | CMV/pAAV | TEVcs(L): ENLYFQ**L**<br>FLAG: DYKDDDDK<br>V5: GKPIPNPLLGLDST<br>TANGO construct, used in *Figure 4—figure supplement 1* |
| P28 | β2AR-V2-TEVcs(L)-FLAG-GAL4-V5 | CMV/pAAV | TEVcs(L): ENLYFQ**L**<br>FLAG: DYKDDDDK<br>V5: GKPIPNPLLGLDST<br>V2: GRTPPSLGPQDESCTTASSSLAKDTSS<br>TANGO construct, used in *Figure 4—figure supplement 1* |
| P29 | Rat-β-arrestin2-HA-TEVp (full length) | CMV/pLX208 | Hygromycin selection marker<br>HAx2: YPYDVPDYAYPYDVPDYA<br>TANGO protease construct for stable integration, used in *Figure 4—figure supplement 1* |
| P30 | β2AR-N-TEVp-iLID-TEVcs(G)-FLAG-GAL4-V5 | CMV/pAAV | TEVcs(G): ENLYFQ**G**<br>FLAG: DYKDDDDK<br>V5: GKPIPNPLLGLDST<br>iTANGO construct, used in *Figure 4—figure supplement 1* |
| P31 | β2AR-V2-N-TEVp-iLID-TEVcs(G)-FLAG-GAL4-V5 | CMV/pAAV | TEVcs(G): ENLYFQ**G**<br>FLAG: DYKDDDDK<br>V5: GKPIPNPLLGLDST<br>iTANGO construct, used in *Figure 4—figure supplement 1* |
| P32 | Rat-β-arrestin2-CTEVp Δ220–242-P2A-tdTomato | CMV/pAAV | iTANGO construct, used in *Figure 4—figure supplement 1* |
| P33 | Grb2-HA-TEVp Δ220–242 | CMV/pAAV | HAx2: YPYDVPDYAYPYDVPDYA<br>SPARK protease construct, used in *Figure 2B* |
| P34 | V5-CRY2PHR-TEVp Δ220–242 | CMV/pAAV | SPARK protease construct, used in *Figure 2C* |
| P35 | TM(CD4)-CIBN- eLOV-TEVcs(Y)-FLAG-GAL4 | CMV/pAAV | TEVcs(Y): ENLYFQ**Y**<br>FLAG: DYKDDDDK<br>SPARK TF construct, used in *Figure 2C* |

DOI: https://doi.org/10.7554/eLife.30233.010

## HEK cell transfection with PEI max

PEI (polyethyleneimine) was our preferred method to transfect HEK cells because it is less expensive and the protocol is simpler than for lipofectamine (one-step rather than two-step). PEI was used for all imaging experiments shown in this study (except for the first three columns of *Figure 2B*) and for the genetic selection experiment in Figure 4. To perform PEI transfection, we treated HEK cells at 60–90% confluence with homemade PEI max solution (Polysciences catalog no. 24765; polyethylenimine HCl Max in $H_2O$, pH 7.3, 1 mg/mL). For transfection of a single well in a 48-well plate, a mixture of DNA (15 ng of UAS-Citrine plasmid; 5–15 ng of SPARK protease plasmid; and 35 ng of SPARK TF plasmid) was incubated with 0.8 μL PEI max solution (1 mg/mL in $H_2O$) in 10 μL serum-free DMEM media for 15 min at room temperature. 100 μL DMEM (Gibco) supplemented with 10% FBS (Sigma) was then mixed with the DNA-PEI max solution and added to the HEK293T cells. Cells were incubated at 37 °C under 5% $CO_2$ until further processing (typically, 15-18 hours). For the DNA quantities, we found that it was important to use a low quantity of SPARK protease DNA (5–15 ng/well in a 48-well plate) in order to achieve low background.

## HEK cell transfection with lipofectamine

For the luciferase platereader assays described in this study, we used lipofectamine to transfect HEK cells instead of PEI. The reason is that for this assay, we had to lift and further divide the cells into multiwell plates ~3 hr following transfection, and this is not compatible with the PEI protocol (which requires the PEI-DNA mixture to remain on the cells for >12 hours) whereas it is compatible with the lipofectamine transfection protocol, described below. HEK293T cells were transfected at 60–90% confluence with Lipofectamine 2000 Transfection Reagent (Invitrogen). For transfection of a single well in a 6-well plate, a mixture of DNA (150 ng of the UAS-reporter plasmid; 100 ng of SPARK protease plasmid; and 350 ng of SPARK TF plasmid) was incubated with 8 μL Lipofectamine in 100 μL serum-free DMEM media for 15 min at room temperature. The media from the well was aspirated and replaced with the DNA-lipofectamine mixture, combined with 2 mL serum-free DMEM. The cells were incubated for 3 h at 37 °C under 5% $CO_2$. Then, the cells were lifted using 400 μL trypsin followed by addition of 1.6 mL complete growth media (DMEM supplemented with 10% FBS and 1% (v/v) Pennicillin-Streptomycin) and replated into 96-well plates. The cells were wrapped in aluminum foil to prevent unwanted activation of SPARK by ambient room light and incubated at 37 °C under 5% $CO_2$ until further processing (typically, 9-12 hours).

## Lentivirus production

HEK293T cells were transfected at 60–90% confluence. For a T25 flask, 2.5 μg viral DNA, 0.25 μg pVSVG, and 2.25 μg delta8.9 lentiviral helper plasmid were combined with 200 μL serum-free DMEM and 30 μL PEI max (Polysciences, Cat# 24765, polyethylenimine HCl Max in $H_2O$, pH 7.3, 1 mg/mL), and incubated for 15 min at room temperature. Then 5 mL DMEM supplemented with 10% FBS was added and mixed. Media was aspirated from the HEK293T cells and the 5 mL DNA mix was added gently to the cells. HEK293T cells were incubated for 48 hr at 37°C and then the supernatant (containing secreted lentivirus) was collected and filtered through a 0.45 μm syringe filter (VWR). Collected lentivirus was divided into aliquots in 0.5 mL sterile eppendorf tubes, flash frozen in liquid nitrogen, and stored at −80°C.

## Stable cell generation

HEK cells were plated on a 6-well plate at ~70–90% confluence. Lentivirus generated from rat-β-arrestin2-TEVp Δ220–242, rat-β-arrestin2-TEVp (full length), or CaM-TEVp Δ220–242 were added to the HEK cells and incubated for 1 day at 37°C under 5% $CO_2$. The media was replaced with fresh complete media, DMEM (Dublecco's Modified Eagle medium, Gibco) supplemented with 10% FBS (Fetal Bovine Serum, Sigma) and 1% (v/v) Penicillin-Streptomycin (VWR, 5000 units/ml of penicillin and 5,000 μg/mL streptomycin), supplemented with 150 ng/mL hygromycin (Corning). From this point on, the media was replaced with fresh complete media supplemented with 150 ng/mL hygromycin (Corning) every day. When cells reached 80% confluence, they were lifted with trypsin and re-plated into a T25 flask and grown until 80% confluence. After one week of hygromycin selection, the stable cells were ready for use, as >95% of the cells grew with continuous treatment of hygromysin.

To confirm the expression of the TEVp construct, we also attempted to do immunostaining of the internal HA epitope tag. However, the immunostaining was very poor, possibly due to steric hindrance of the internal HA epitope tag. Therefore, we tested the expression of the TEV protease by SPARK, as described in section 'HEK cell stimulation, fixation, and immunostaining'.

## HEK cell stimulation, fixation, and immunostaining

It is critical to stimulate the HEK cells **15–18** hr post-transfection and fix **7–9** hr post-stimulation to avoid background accumulation. To stimulate the cells with drug, 100 μL of drug in complete growth media, DMEM (Gibco) supplemented with 10% FBS (Sigma) and 1% (v/v) Penicillin-Streptomycin (VWR, 5000 units/ml of penicillin and 5,000 μg/mL streptomycin), was added gently to the top of the media in a 48-well plate to the final concentrations indicated. For no-drug conditions, 100 μL complete growth media was added. For light stimulation, we used either a home-made LED light array (467 nm, 60 mW/cm$^2$, 10–33% duty cycle, 2 s of light every 6 s) at 37°C, ambient room light, or a daylight lamp (T5 Circline Fluorescent Lamp, 25W, 6500K, 480 nm/530 nm/590 nm) at 25°C. When using ambient room light or a daylight lamp, we kept the aluminum foil underneath the 96-well plate for better reflection of light to the cells. After the stimulation period, the solution in the 48-well plates was removed and the cells were washed once with complete growth media and then incubated with 200 μL of complete growth media. Thereafter, HEK cells were incubated for 9 hr at 37°C before fixation with 4% paraformaldehyde in PBS (phosphate buffered saline) for 15 min at room temperature. HEK293T cells were permeabilized by incubation with cold methanol at −20°C for 5 min, then immunostained with mouse-anti-V5 antibody (1:2000 dilution, Life Technology, *Table 2*), chicken anti-myc (1:1000 dilution, Life Technology) and/or rabbit-anti-HA antibody (1:1000 dilution, Rockland) in 2% BSA solution in PBS for 30 min at room temperature with gentle rocking. The cells were washed twice with room temperature PBS and then incubated with secondary antibodies: anti-mouse-Alexa Fluor 405, anti-chicken-Alexa Fluor 647, and anti-rabbit-Alexa Fluor 568 (1:1000 dilution for each, Life Technology) in 2% BSA solution in PBS, for 20 min at room temperature. Cells were washed twice with PBS and maintained in PBS at 4°C until imaging. HEK 293T cells were imaged with the 10x air objective on the Zeiss AxioObserver confocal microscope, as described above. Eight to ten fields of view were acquired for each condition.

## Quantification of EGFR-Grb2 SPARK images

Quantification of EGFR-Grb2 SPARK images (*Figure 2B*) was performed using Slidebook 6 (Intelligent Imaging Innovations). A mask was created to capture all cells with Citrine expression. The sum of the Citrine intensity inside the mask was calculated, then background-corrected using an ROI

**Table 2.** Antibodies used for immunofluorescence.

| Antibody | Source | Company | Catalog number | Dilutions used |
|---|---|---|---|---|
| anti-Flag | Mouse | Sigma | F3165-1MG | HEK293T immunostaining: 1:1000 |
| anti-V5 | Mouse | Life Technologies | R96025 | HEK293T immunostaining: 1:2000 |
| anti-HA | Rabbit | Rockland | 600-401-384 | HEK293T immunostaining: 1:1000 |
| anti-Myc | Chicken | Life Technologies | A-21281 | HEK293T immunostaining: 1:1000 |
| anti-Mouse-AlexaFluor488 | Goat | Life Technologies | A-11001 | HEK293T immunostaining: 1:1000 |
| anti-Mouse-AlexaFluor568 | Goat | Life Technologies | A-11004 | HEK293T immunostaining: 1:1000 |
| anti-rabbit-AlexaFluor568 | Goat | Life Technologies | | HEK293T immunostaining: 1:1000 |
| anti-Mouse-AlexaFluor647 | Goat | Life Technologies | A-21235 | HEK293T immunostaining: 1:1000 |
| anti-chicken-647 | Goat | Life Technologies | | HEK293T immunostaining: 1:1000 |
| anti-mouse-HRP | Goat | Bio-Rad | | Western blot: 1:2000 |

DOI: https://doi.org/10.7554/eLife.30233.011

outside the Citrine-positive cells. Five separate fields of view were analyzed per condition, and the mean intensity was used to calculate the signal ratios.

## Luciferase assays

HEK293T cells in 6-well plates were transfected with 150 ng UAS-luciferase, 350 ng SPARK TF component, 100 ng SPARK protease component and 8 µL lipofectamine in FBS-free DMEM media for 3 hr at 37°C under 5% $CO_2$. Alternatively, HEK cells stably expressing arrestin-TEVp in 6-well plates were transfected with 150 ng UAS-luciferase, 350 ng SPARK TF component and 8 µL lipofectamine in FBS-free DMEM media for 3 hr at 37°C under 5% $CO_2$. The transfection media was then removed and complete growth media, DMEM (Dublecco's Modified Eagle medium, Gibco) supplemented with 10% FBS (Fetal Bovine Serum, Sigma) and 1% (v/v) Penicillin-Streptomycin (VWR, 5000 units/ml of penicillin and 5,000 µg/mL streptomycin), was added to each well and the cells were kept in dark by wrapping the plates in aluminum foil. Plates were incubated at 37°C under 5% $CO_2$. Subsequent procedures were performed in a dark room with red light illumination. After 3 hr of incubation, the cells from each well were lifted with 300 µL trypsin at 25°C for 2–3 min, and 1.7 mL complete growth media was added and pipetted to create cell suspensions. 100 µL of the cell suspension was added to each well of a flat, white-wall, clear bottom 96-well plate (Greiner Bio-One) that was pretreated with 50 µg/mL human fibronectin (EMD Millipore) by first incubating at 37°C for 20 min and then removed. Cells were allowed to settle for an additional 9–12 hr before stimulation.

For light stimulation, we used either a home-made LED light array (467 nm, 60 mW/cm$^2$, 10–33% duty cycle, 2 s of light every 6 s) at 37°C, ambient room light or a daylight lamp (T5 Circline Fluorescent Lamp, 25W, 6500K, 480 nm/530 nm/590 nm) at 25°C. When using ambient room light or a daylight lamp, we kept the aluminum foil underneath the 96-well plate for better reflection of light to the cells. For drug stimulation, we added 100 µL of complete growth media with the ligand to the final concentration indicated. For conditions without drug stimulation, we simply added 100 µL of complete growth media. After stimulation, the cells were rewrapped in aluminum foil and incubated at 37°C under 5% $CO_2$ for 9 hr until the luciferase assay. <u>Important note</u>: *We found **7–9 hr** incubation after stimulation (for transcription and translation of the UAS reporter gene) to be optimal for SPARK in HEK cells. Longer incubation times post-stimulation, such as 24 hr, resulted in lower signal ratios due to higher background.*

For the luciferase assay, we used the Bright-Glo luciferase assay system (Promega). The Bright-Glo reagent was prepared in 0.8 mL aliquots and stored at −20°C. 30 min prior to the luciferase assay, the Bright-Glo reagent was thawed in a water bath at 25°C to ensure the equilibration of the reagent to 25°C prior to usage.

Media was then aspirated from each well in the 96-well plate and 100 µL DPBS was added to each well followed by 100 µL of the Bright-Glo reagent. The cells were immediately analyzed using a plate reader (Tecan Infinite M1000 Pro) using the following parameters: 1000 msec acquisition time, green-1 filter (520–570 nm), 25°C, linear shaking for 3 s.

## SPARK applied to various GPCRs (*Figure 2D*)

Following the procedures in the above section, 'Luciferase assays', HEK cells stably expressing β-arrestin2-TEVp Δ220–242 were initially plated into 6-well plates and transfected with UAS-Luciferase (150 ng, plasmid 2 in *Table 1*), GPCR-LOV-TEVcs-GAL4 (350 ng, plasmid 6, 20, 16, 17, 19, 15, 13 or 14 in *Table 1*) at 50–80% confluence with lipofectamine. 6 hr after transfection, HEK cells were replated to 96-well plates in the dark, wrapped in aluminum foil and incubated at 37°C under 5% $CO_2$ for another 12 hr. HEK cells were stimulated with daylight lamp and corresponding drug (10 µM) for 15 min, incubated for another 9 hr at 37°C under 5% $CO_2$ and then assayed using Bright-Glo luciferase assay system (Promega) as described above.

## Comparison of LOV domains in HEK cells stably expressing arrestin2-TEVp (*Figure 1—figure supplement 1C*)

Following the procedures in the above section, 'Luciferase assays', HEK cells stably expressing β-arrestin2-TEVp Δ220–242 were initially plated into 6-well plates and transfected with UAS-Luciferase (150 ng, plasmid 2 in *Table 1*), β2AR-LOV-TEVcs-GAL4 (350 ng, plasmid 6, 21, 22, 23, or 24 in *Table 1*) at 50–80% confluence with lipofectamine. 6 hr after transfection, HEK cells were replated to

96-well plates in the dark, wrapped in aluminum foil and incubated at 37°C under 5% $CO_2$ for another 12 hr. HEK cells were stimulated with daylight lamp and isoproterenol (10 uM) for 15 min, incubated for another 9 hr at 37°C under 5% $CO_2$ and then assayed using Bright-Glo luciferase assay system (Promega) as described above.

### Comparison of LOV domains in HEK cells transiently expressing arrestin2-TEVp (*Figure 1—figure supplement 1D and E*)

Following the procedures in the above section, 'Luciferase assays', HEK cells were initially plated in 6-well plate and transfected with UAS-Luciferase (150 ng, plasmid 2 in *Table 1*), β2AR-LOV-TEVcs-GAL4 (350 ng, plasmid 6, 21, 22, 23, or 24 in *Table 1*) and arrestin-TEVp Δ220–242 (100 ng, plasmid 4 in *Table 1*) at 50–80% confluence with lipofectamine. 6 hr after transfection, HEK cells were replated to 96-well plate in dark, wrapped in aluminum foil and incubated at 37°C under 5% $CO_2$ for another 12 hr. HEK cells were stimulated with daylight lamp and isoproterenol (10 μM) for 5 or 15 min, incubated for another 9 hr at 37°C under 5% $CO_2$ and then assayed using Bright-Glo luciferase assay system (Promega).

### Western blot to determine SPARK cleavage extent (*Figure 1—figure supplement 2C*)

HEK cells plated in a 6-well plate were transfected at 60–80% confluence with β2AR-eLOV-TEVcs-GAL4 (500 ng, plasmid 6 in *Table 1*) and arrestin-TEVp Δ220–242 (150 ng, plasmid 4 in *Table 1*) using PEI max. 18 hr post-transfection, HEK cells were stimulated with or without isoproterenol (10 μM) in light (467 nm, 60 mW/cm$^2$, 10% duty cycle, 0.5 s of light every 5 s) or in dark for 5 min or 30 min. HEK cells were washed once with freshly-made iodoacetamide (10 mM in dPBS) immediately after stimulation, in order to inhibit TEVp activity, then cells were harvested and pelleted in the dark. Cell pellets were resuspended in RIPA buffer (50 mM Tris pH 7.5, 150 mM NaCl, 0.1% SDS, 0.5% sodium deoxycholate, and 1% Triton X-100) supplemented with 10 mM iodoacetamide and 1% protease inhibitor cocktail (PIC, Sigma Aldrich, catalog no. P8849), then clarified by centrifugation at 15,000 rpm at 4°C. Cleared lysates were combined with protein loading buffer, boiled for 2 min, and then separated on a 8% SDS-PAGE gel. After transfer to a nitrocellulose membrane, SPARK TF component was detected by blotting with mouse anti-V5 antibody. To perform the blotting, the membrane was gently rocked at 25°C in 2% w/v bovine serum albumin (BSA, Thermo Fisher) for 1 hr. The membrane was immersed in 1:1000 mouse anti-V5 (Life Technologies) in 2% BSA in 1 × TBST (0.1% Tween-20 in Tris-buffered saline) for 1 hr at 25°C with gentle rocking. The membrane was rinsed with 1 × TBST three times and then the membrane was incubated with goat anti-mouse-HRP (Bio-Rad, 1:2000 dilution) in 2% w/v BSA in 1 × TBST for one hour at room temperature. The membrane was rinsed with 1 × TBST three times, then developed with SuperSignal West Pico reagent (Thermo Scientific) and imaged on an Alpha Innotech gel imaging system. We used ImageJ to quantify band intensities.

### Comparison of SPARK to TANGO (*Figure 4—figure supplement 1B*)

Following the procedures in the section 'Luciferase assay', HEK cells were initially plated in 6-well plate. For SPARK, HEK cells stably expressing arrestin-TEVΔ220–242 were transfected with UAS-Luciferase (150 ng, plasmid 2 in *Table 1*), β2AR-eLOV-TEVcs-GAL4 (350 ng, plasmid 6 or 26 in *Table 1*); for TANGO, HEK cells stably expressing arrestin-TEVΔ220–242 were transfected with UAS-Luciferase (150 ng, plasmid 2 in *Table 1*), β2AR-TEVcs-GAL4 (350 ng, plasmid 27 or 28 in *Table 1*) at 50–80% confluence with lipofectamine. 6 hr after transfection, HEK cells were replated into a 96-well plate in the dark, wrapped in aluminum foil and incubated at 37°C under 5% $CO_2$ for another 12 hr until stimulation. For stimulation, we used a daylight lamp and isoproterenol (final concentration 10 μM) for the indicated times. For 15 min light and isoproterenol treatment, the cells were incubated for another 9 hr at 37°C after stimulation under 5% $CO_2$ and then luciferase expression was quantified on a platereader as described above ('Luciferase assay'). For 15 min light and 18 hr isoproterenol treatment, the cells were incubated for another 17 hr and 45 min at 37°C after stimulation under 5% $CO_2$ and then luciferase expression was quantified on a platereader as described above ('Luciferase assay').

## Comparison of SPARK to iTANGO (*Figure 4—figure supplement 1C*)

Following the procedures in the section 'Luciferase assay', HEK cells were initially plated in 6-well plate. For SPARK, HEK cells were transfected with UAS-Luciferase (150 ng, plasmid #2 in *Table 1*), β2AR-(±V2)-eLOV-TEVcs-GAL4 (350 ng, plasmid #6 or 26 in *Table 1*) and arrestin-TEVp Δ220–242 (100 ng plasmid #4 in *Table 1*) at 50–80% confluence with lipofectamine; for iTANGO, HEK cells were transfected with UAS-Luciferase (150 ng, plasmid #2 in *Table 1*), β2AR-(±V2)-NTEVp-iLOV-TEVcs-GAL4 (350 ng, plasmid #30 or 31 in *Table 1*) and arrestin-CTEVp-P2A-tdTomato (100 ng plasmid #32 in *Table 1*) at 50–80% confluence with lipofectamine. 6 hr after transfection, HEK cells were replated into a 96-well plate in the dark, wrapped in aluminum foil and incubated at 37°C under 5% $CO_2$ for another 12 hr. For light stimulation, we used a daylight lamp for the indicated times. For no light condition, the cells were wrapped in aluminum foil. For drug stimulation, isoproterenol (final concentration 10 µM) was added during the stimulation and remained in the wells until the end of the experiment. After stimulation, the cells were incubated for another 9 hr at 37°C under 5% $CO_2$ and then luciferase expression was quantified on a platereader as described above ('Luciferase assay').

## Time course assays (*Figure 3C, D and E*)

Following the procedures in the section 'Luciferase assay', HEK cells stably expressing Rat-β-arrestin2-HA-TEVp Δ220–242 were initially plated in 6-well plates and transiently transfected with SPARK constructs, UAS-Luciferase (150 ng, plasmid 2 in *Table 1*) and SPARK-TF construct (350 ng) using lipofectamine for 3 hr. The cells were lifted using trypsin (Invitrogen) 3 hr post-transfection and replated in 96-well plate pretreated with human fibronectin. 15 hr post-transfection, corresponding ligands were added to the HEK cells to a final concentration of 10 µM and the ligands were kept in the media until the end of the experiment for *Figure 3C and E*. For *Figure 3D*, ligands were added to the cells fro the first 5 min and immediately removed and washed once with complete growth media, DMEM (Gibco) supplemented with 10% FBS (Sigma) and 1% (v/v) Penicillin-Streptomycin (VWR, 5000 units/ml of penicillin and 5,000 µg/mL streptomycin). Light (Daylight lamp, 5 min) was used to stimulate the cells at various time points post-ligand addition as indicated in *Figure 3C and E*. 9 hr post-stimulation, HEK cells were assayed using Bright-Glo luciferase assay system (Promega).

## SPARK applied to various different PPIs (*Figure 2B*)

HEK cells were plated in 48-well plates that were pretreated with 50 µg/mL human fibronectin (Millipore) for at least 10 min at 37°C before cell plating (to improve adherence of HEK cells). For DRD1, NMBR, EGFR, HEK cells were transiently transfected with UAS-Citrine (15 ng, plasmid # one in *Table 1*), SPARK TF component (35 ng, plasmid # 16, 18, or 19 in *Table 1*) and SPARK protease component (15 ng, plasmid # 5 or 33 in *Table 1*) using lipofectamine (see methods section 'HEK cell transfection with lipofectamine'). For CD4-FRB-FKBP and mito-FRB-FKBP, HEK cells were transiently transfected with UAS-Citrine (15 ng, plasmid # one in *Table 1*), SPARK TF component (35 ng) and SPARK protease component (15 ng) using PEI max (see methods section 'HEK cell transfection with PEI max').

For EGFR stimulation, HEK cell were kept in FBS free DMEM media for 18 hr before treatment with EGF (Corning, 10 ng/mL) and light (daylight lamp) for 5 min. NMBR-arrestin SPARK was stimulated with bombesin (Sigma, 10 µM) and light (daylight lamp) for 5 min. DRD1-arrestin SPARK was stimulated with dopamine (Sigma, 10 µM) and light (daylight lamp) for 5 min. FRB-FKBP SPARK were stimulated with rapamycin (Alfa Aesar, 100 nM) and light (blue LED light array) for 5 min.

9 hr after stimulation, HEK cells were fixed and imaged with the 10x air objective on the Zeiss AxioObserver inverted confocal microscope. Five fields of view were acquired for each condition.

## SPARK for detection of CRY-CIBN interaction (*Figure 2C*)

HEK cells were plated in 48-well plates that were pretreated with 50 µg/mL human fibronectin (Millipore) for at least 10 min at 37°C before cell plating (to improve adherence of HEK cells). HEK cells were transfected with SPARK TF component (50 ng, plasmid 35 in *Table 1*), SPARK protease component (30 ng, plasmid 34 in *Table 1*) with PEI max. At the same time, lentivirus encoding UAS-Citrine (20 uL) was added to the HEK cells in each well. The cells were wrapped in aluminum foil. 18 hr post-transfection, the HEK cells were stimulated by exposing to light (467 nm, 60 mW/cm², 33% duty

cycle, 2 s of light every 6 s) for 2 min, 4 min and 8 min or kept in dark. 9 hr post-stimulation, HEK cells were fixed and imaged.

## SPARK proximity experiment (Figure 1F)

HEK cells were plated directly in plastic 48-well plates that were pretreated with 50 µg/mL human fibronectin (Millipore) for at least 10 min at 37°C before cell plating (to improve adherence of HEK cells). HEK cells were transfected with SPARK reporter component UAS-Citrine (15 ng, plasmid # one in Table 1), SPARK TF component (35 ng, plasmid # 6 or eight in Table 1), SPARK protease component (5 ng, plasmid # 3 or seven in Table 1) and with or without HA-β2AR (35 ng, plasmid # 6 or eight in Table 1) with 0.8 uL PEI max in complete growth media, DMEM (Gibco) supplemented with 10% FBS (Sigma) and 1% (v/v) Penicillin-Streptomycin (VWR, 5000 units/ml of penicillin and 5,000 µg/mL streptomycin), in dark for 15 hr at 37°C under 5% $CO_2$. HEK cells were treated with light (daylight lamp) and isoproterenol (10 uM) for 5 min and then incubated for another 9 hr at 37°C under 5% $CO_2$. HEK cells were then fixed with 4% formaldehyde, immunostained and imaged as described in section 'HEK cell stimulation, immunofluorescence and imaging'. We used primary antibodies mouse-anti-V5 (1:2000 dilution, Life Technology), chicken anti-myc (1:1000 dilution, Life Technology) and rabbit-anti-HA (1:1000 dilution, Rockland) and secondary antibodies anti-mouse-Alexa Fluor 405 (1:1000 dilution, Life Technology), anti-chicken-Alexa Fluor647 and anti-rabbit-Alexa Fluor 568 (1:1000 dilution, Life Technology). Stained HEK 293T cells were imaged with the 10x air objective on the Zeiss AxioObserver inverted confocal microscope. Five fields of view were acquired for each condition.

## Live cell imaging of rat-β-arrestin2-GFP translocation (Figure 1—figure supplement 2A)

HEK293T cells were plated on glass coverslips pretreated with human fibronectin (Millipore), so that it reaches 60–80% confluence next day. The following day, HEK293T cells from each well were transfected with rat-β-arrestin2-GFP (10 ng) and a transmembrane construct (35 ng of HA-β2AR, β2AR SPARK TF component or TM(CD4) SPARK TF component) with PEI max at 37°C under 5% CO2 for 18 hr. The live cells were transferred to an imaging dish filled with DPBS (Dulbecco's Phosphate Buffered Saline) with the cells facing up and imaged with the 63x oil objective on the Zeiss AxioObserver inverted confocal microscope at room temperature. For the same field of view, images were taken before and 5 min after isoproterenol treatment. For isoproterenol treatment, 100 µL isoproterenol in DPBS was gently added to the top of the coverslips to a final concentration of 10 µM.

## Cell based selection using SPARK (Figure 4)

HEK cells stably expressing arrestin-TEVp Δ220–242 and CaM-TEVp Δ220–242 were co-cultured at a ratio of 1:50 in a T75 flask in 15 mL complete media, DMEM (Dublecco's Modified Eagle medium, Gibco) supplemented with 10% FBS (Fetal Bovine Serum, Sigma) and 1% (v/v) Penicillin-Streptomycin (VWR, 5000 units/ml of penicillin and 5,000 µg/mL streptomycin), at 37°C under 5% $CO_2$. At 40–60% confluence, the cells were transfected with UAS-citrine (1,125 ng, plasmid # 1 in Table 1) and β2AR-eLOV-TEVcs-FLAG-GAL4-V5 (2265 ng, plasmid # 6 in Table 1) using 60 µL PEI max, and the cells were wrapped in aluminum foil to prevent unwanted activation of the LOV domain by ambient room light. 18 hr post-transfection, the cells were stimulated with light (10 min, daylight lamp) and isoproterenol (10 µM final concentration) at 25°C. After stimulation, the cells were washed once and then further incubated with 15 mL complete media for 9 hr in the dark at 37°C under 5% $CO_2$. The cells were then lifted with 3 mL trypsin (Invitrogen) and resuspended in 12 mL FACS buffer (ice cold RPMI buffer supplemented with 2% FBS (Fetal Bovine Serum, Sigma), 1% (v/v) Penicillin-Streptomycin (VWR, 5000 units/ml of penicillin and 5000 mg/mL streptomycin), and 20 mM HEPES buffer (Thermofisher)). 15 µL of the cells were pelleted and frozen at -80°C freezer right away for 'pre-sort' analysis. The rest of the cells were pelleted by centrifugation at 1000 r.p.m for 3 min at RT. The pellet was washed once with the FACS buffer and then resuspended in 3 mL FACS buffer to a final concentration of roughly $5 \times 10^6$ cells/mL. The cell suspension was filtered through the cell-strainer cap on the polystyrene tubes (FALCON, CAT# 352235) into the 5 mL polystyrene tubes (FALCON) and kept on ice until sorting.

FACS was performed on a SONY sorter (SH800S, 488 nm laser, optical filter 525/50), and a sorting gate was drawn to capture all the cells above the threshold of the CaM-TEV cells (determined by separate FACS analysis of a control sample). Before sorting, HEK cells were pipetted up and down five times and filtered again into the polystyrene tubes, to ensure complete separation between single cells and removal of cell clumps. During sorting, the source cells were pipetted up and down every 10 min to prevent aggregate formation. We tried addition of EDTA (1–5 mM) to prevent cell aggregation; however, cell recovery was poor (<5% cells survived). Therefore, we used pipetting every 10 min during sorting instead to break up aggregates. In total, around 1000 cells (0.01% of total cells) were collected. Cells were sorted into a 15 mL conical tube with 4 mL FACS collection media (DMEM supplemented with 30% FBS (Sigma), 1% (v/v) Penicillin-Streptomycin (VWR, 5000 units/ml of penicillin and 5,000 μg/mL streptomycin) and 20 mM HEPES buffer (Thermofisher)).

The collected cells were immediately put on ice and pelleted by centrifugation at 1000x RPM at 25 °C for 3 min using eppendorf Centrifuge 5810 R. The media was gently removed. The cell pellet (not visible) was resuspended with 400 μL FACS collection media and transferred to a 1.6 mL eppendorf tube. The cells were pelleted again by centrifugation at 1000x g.

## RNA extraction from HEK cells and RNA reverse transcription

RNA was extracted from the pelleted cells from above following RNeasy Plus Mini Kit (QIAGEN). The cell pellet was resuspended in 350 μL RLT Plus buffer with 40 μM DTT and homogenized by pipetting up and down at least 10 times. The lysate was transferred to a gDNA Eliminator spin column and centrifuged at >8000 x g for 1 min. 350 μL of 70% ethanol was added to the flow through, mixed and transferred to an RNeasy spin column. The bound RNA was washed with 700 μL RW1 once, 500 μL RPE once and then eluted with 30 μL RNase-free water supplemented with 1 μL RNase inhibitor (Ribolock, Thermo Fisher).

Extracted RNA was reverse transcribed using the SuperScript III Reverse Transcriptase kit (Thermo Fisher), priming with random hexamers (Thermofisher) according to the manufacturer's protocol.

## qPCR quantification of reverse transcribed RNA

The ratio of arrestin-TEVp: CaM-TEVp in the cell-extracted RNA was analyzed by qPCR. For each qPCR reaction, we mixed 5 μL 2X FastStart Universal SYBR Green Master mix (Roche), 3 μM of forward and reverse primer for arrestin or CaM (sequences shown below), 2 μL of the reverse transcribed RNA and ddH$_2$O to a final volume of 10 μL.

Arrestin forward primer: 5' CTCTCTGGCTAACTGTCGGG
Arrestin reverse primer: 5' TACACGGTGAGCTTGCAGTT
CaM forward primer: 5' TCTCTGGCTAACTGTCGGGA
CaM reverse primer: 5' ATCCCCGTCCTTGTCCAGTA

We used Applied Biosystem StepOne qPCR instrument and followed the standard Sybergreen protocol but with 55 amplification cycles to reach the sybergreen signal plateau. Three technical replicates were performed for each condition. Raw data was analyzed to calculate Ct number and qPCR efficiency, ε, using Real time qPCR Miner: http://ewindup.info/miner/data_submit.htm

Relative amounts of each gene were calculated using the following equation:

RNA amount = $1/(1 + \varepsilon)^{Ct}$

Enrichment factor = $(1 + \varepsilon_{calmodulin})^{Ct\ post\text{-}sorting\ -\ Ct\ pre\text{-}sorting} / (1 + \varepsilon_{arrestin})^{Ct\ post\text{-}sorting\ -\ Ct\ pre\text{-}sorting}$

Error bar = $(((1 + \varepsilon_{CaM})^{Ct\ STD} - 1)^2 + ((1 + \varepsilon_{arrestin})^{Ct\ STD} - 1)^2)^{1/2}$

## Acknowledgements

We thank K K Kumar and B Kobilka for helpful discussions. FACS experiments were performed in the Stanford Shared FACS Facility. Rat β-arrestin2 and β2AR genes were gifts from R. Lefkowitz, Duke University. The EGFR gene was a gift from M Meyerson, Dana-Farber Cancer Institute. The Grb2 gene was a gift from B Mayer, University of Connecticut. The luciferase gene was a gift from M Sato, University of Tokyo. DRD1, DRD2, NMBR, MLNR, AVPR2, AGTR2, and ADRA1A genes were gifts from B Roth, University of North Carolina. AYT received funding from Stanford University and is a Chan Zuckerberg Biohub investigator.

# Additional information

## Competing interests
Wenjing Wang, Alice Y Ting: AYT and WW have filed a patent application covering some aspects of this work.(U.S. Pat. App. No. 62/440,825). The other authors declare that no competing interests exist.

## Funding

| Funder | Grant reference number | Author |
|---|---|---|
| National Institutes of Health | DA010711 | Mark von Zastrow |
| National Institutes of Health | DA012864 | Mark von Zastrow |
| Stanford University | | Alice Y Ting |

The funders had no role in study design, data collection and interpretation, or the decision to submit the work for publication.

## Author contributions
Min Woo Kim, Data curation, Formal analysis, Validation, Investigation, Visualization, Methodology, Writing—original draft, Writing—review and editing; Wenjing Wang, Conceptualization, Data curation, Formal analysis, Validation, Investigation, Visualization, Methodology, Writing—original draft, Writing—review and editing; Mateo I Sanchez, Mark von Zastrow, Robert Coukos, Methodology, Writing—review and editing; Mark von Zastrow, Writing—review and editing; Alice Y Ting, Conceptualization, Supervision, Funding acquisition, Visualization, Methodology, Writing—original draft, Project administration, Writing—review and editing

## Author ORCIDs
Min Woo Kim  https://orcid.org/0000-0003-1327-3617
Wenjing Wang  https://orcid.org/0000-0001-6025-9848
Mateo I Sanchez  https://orcid.org/0000-0003-1359-6969
Robert Coukos  https://orcid.org/0000-0002-7307-8293
Mark von Zastrow  https://orcid.org/0000-0003-1375-6926
Alice Y Ting  https://orcid.org/0000-0002-8277-5226

## Decision letter and Author response
Decision letter https://doi.org/10.7554/eLife.30233.014
Author response https://doi.org/10.7554/eLife.30233.015

# Additional files

## Supplementary files
• Transparent reporting form
DOI: https://doi.org/10.7554/eLife.30233.012

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
