## [Decision Letter]

Thank you for submitting your article "Time-gated detection of protein-protein interactions with transcriptional readout" for consideration by *eLife*, which was considered as a Tool and Resources article. Your article has been favorably evaluated by Philip Cole (Senior Editor) and two reviewers, one of whom, Volker Dötsch (Reviewer #1), is a member of our Board of Reviewing Editors.

The reviewers have discussed the reviews with one another and the Reviewing Editor has drafted this decision to help you prepare a revised submission.

Summary:

The authors describe the engineering and testing in cell culture of a system (TIGER) for reporting protein-protein interactions within a time window gated by blue light (time resolution ~5min) as transcriptional activation of a reporter gene.

The current work directly compares the performance of the TIGER system with the previously published iTango, since both focus primarily on GPCR-arrestin interactions. TIGER (current work) has better contrast +/- light and +/- agonist, primarily as a result of lower background, relative to iTANGO. It also has better kinetics which provide improved time resolution, both of which are beneficial in this approach. The current work attempts to generalize by showing several additional protein-protein interactions, including EGFR-Grb2, FRB-FKBP, and CIBN-CRY2PHR.

Overall, the work successfully improves upon molecular tools for driving transcription in response to light + protein+protein interactions, an approach which will likely prove valuable for many aspects of cell and systems biology. The approach appears general at some level across protein-protein interactions, although for some interactions the contrast is dramatically reduced and may require optimization.

Essential revisions:

1) Unfortunately, there is no quantification of experiments describing non membrane protein interactions, only low-resolution images of cell culture. Figure 2 shows that the efficacy of the approach varies widely depending on the specific protein-protein interaction being studied. A quantification of results from these protein-protein interactions (non-GPCR-arrestin) is required.

2) The method proposed is conceptually very similar to three systems published earlier this year:

Cal-Light:https://www.ncbi.nlm.nih.gov/pubmed/28650460

FLARE:https://www.ncbi.nlm.nih.gov/pubmed/28650461iTango:https://www.ncbi.nlm.nih.gov/pubmed/28369042

Despite the similarities with previously reported systems, there is no mention of these in the Introduction of this manuscript, where comparisons are drawn between TIGER and other molecular approaches for monitoring protein-protein interactions. These other techniques should be mentioned in the Introduction and similarities and differences to these systems should be discussed.

---

## [Author Response]

Essential revisions:1) Unfortunately, there is no quantification of experiments describing non membrane protein interactions, only low-resolution images of cell culture. Figure 2 shows that the efficacy of the approach varies widely depending on the specific protein-protein interaction being studied. A quantification of results from these protein-protein interactions (non-GPCR-arrestin) is required.

We now have quantification of all PPIs shown in Figure 2:

· DRD1-Arrestin PPI quantification (luciferase readout) shown in Figure 2;

· NMDR-Arrestin PPI quantification (luciferase readout) shown in Figure 2;

· EGFR-Grb2 PPI quantification (Citrine readout) given in the legend of Figure 2;

· FRB-FKBP PPI quantification (luciferase readout) shown in Figure 2—figure supplement 1;

· Mito-FRB/FKBP PPI quantification (luciferase readout) shown in Figure 2—figure supplement 1;

· CIBN-CRY2PHR PPI quantification (luciferase readout) shown in Figure 2—figure supplement 1.

2) The method proposed is conceptually very similar to three systems published earlier this year:Cal-Light:https://www.ncbi.nlm.nih.gov/pubmed/28650460FLARE:https://www.ncbi.nlm.nih.gov/pubmed/28650461iTango:https://www.ncbi.nlm.nih.gov/pubmed/28369042Despite the similarities with previously reported systems, there is no mention of these in the Introduction of this manuscript, where comparisons are drawn between TIGER and other molecular approaches for monitoring protein-protein interactions. These other techniques should be mentioned in the Introduction and similarities and differences to these systems should be discussed.

Our Introduction and Abstract both mention Tango, which is highly relevant to SPARK (the new name for our tool, changed from TIGER) as it is also a transcriptional PPI tool. We now explicitly mention FLARE and what it is for, as soon as we begin to discuss the design of our SPARK tool, because that is the way that FLARE is relevant – in its design, but not in its purpose (its purpose is to sense calcium, not PPIs). iTango was published as a dopamine sensing tool, not a PPI detection methodology, and therefore we do not believe it is appropriate to mention it in our Introduction of PPI methods. Later in the manuscript, we have an entire section on iTango and a side by side comparison to SPARK (because we extrapolate that iTango *could* be used for PPI detection, even though the previous study did not demonstrate it). Cal-Light is even less relevant, as it is not for PPIs, and its design is different from that of SPARK and FLARE; nevertheless we do cite Cal-Light when we discuss LOV domain optimization.